# Winsorization for Robust Bayesian Neural Networks

**DOI:** 10.3390/e23111546

**Published:** 2021-11-20

**Authors:** Somya Sharma, Snigdhansu Chatterjee

**Affiliations:** 1Department of Computer Science and Engineering, University of Minnesota-Twin Cities, 200 Union Street SE, Minneapolis, MN 55455, USA; sharm636@umn.edu; 2School of Statistics, University of Minnesota-Twin Cities, 313 Ford Hall, 224 Church St. SE, Minneapolis, MN 55455, USA

**Keywords:** Bayesian neural network, uncertainty quantification, variational Gaussian process, Winsorization, concrete dropout, flipout, mixture density networks

## Abstract

With the advent of big data and the popularity of black-box deep learning methods, it is imperative to address the robustness of neural networks to noise and outliers. We propose the use of Winsorization to recover model performances when the data may have outliers and other aberrant observations. We provide a comparative analysis of several probabilistic artificial intelligence and machine learning techniques for supervised learning case studies. Broadly, Winsorization is a versatile technique for accounting for outliers in data. However, different probabilistic machine learning techniques have different levels of efficiency when used on outlier-prone data, with or without Winsorization. We notice that Gaussian processes are extremely vulnerable to outliers, while deep learning techniques in general are more robust.

## 1. Introduction

Machine learning (ML) and artificial intelligence (AI) techniques have met astounding success in different industries and research problems. Conventionally, these techniques have the singular focus of improving prediction accuracy in complex data analysis problems. Despite the mass applicability and popularity of ML prediction methods, many of the related architectures fail to account for the fact that in many large datasets, there are potential outlying observations in both the target variable and the features. Unlike classical statistical frameworks involving relatively small datasets with few features, it is not possible in big data to carefully select and then either drop or modify observations in a pre-processing step prior to the main data analysis. In any case, such ad hoc pre-processing steps can lead to a violation of standard regularity conditions that are required for a proper probabilistic analysis [1,2]. This is essentially a result of using the data twice, once for outlier detection and then again for constructing the predictive model, whereby there is a false sense of accuracy and precision for the second step. Similar issues have been noted in the context of model selection and other problems also, see [3] and related literature for deep theoretical discussions and results.

The problem of outliers in the data is exacerbated when such data are used with deep learning (DL) or related black-box techniques that are supremely versatile. Because of the inherent strengths of these techniques, they may yield excellent numeric summaries such as mean squared errors even on data with outliers by simply overfitting near such aberrant observations. Such aspects of DL fitting have been observed earlier and are of considerable interest in studies on the properties of DL [4,5]. In essence, standard outlier detection techniques such as studies on residuals are not operative owing to localized overfitting by the DL architecture, and the use of robust model fitting procedures are not viable because they scale poorly with data size or parameter size and hence pose extremely burdensome computational requirements.

In this paper, we propose a probabilistic Winsorization step on the training data, to mitigate the adversarial effects of model learning on noisy and outlier-prone data. Consider, for a moment, a numeric dataset on a single variable. In this case, the data can be ordered, say, in an increasing order. Winsorization is a process where the highest and lowest α-fraction of the observations are replaced by their nearest neighbors in the remaining “central” (1−2α)-fraction of the data. For example, if there are 100 observations in an increasing order and if α=0.05, we replace each of the smallest five values with the sixth lowest value and each of the highest five values with the 95-the highest value. Thus, the original data size remains intact, outliers are dropped from the data, human-centric pre-processing of the data is not needed, and complex mathematical formulations and optimizations are not required to ensure robustness. The value of α can be chosen using a trade-off between efficiency and robustness, or some other criteria, and it is trivial to generalize to the case where different fractions of observations are selected to be replaced from the upper and lower tails. A minor generalization is achieved when some random noise is added to each of the 2α replacement observations that are used in place of the highest and lowest actual data or some other systematic transformations used on these. The statistical properties of hyperparameter estimators, predictions, and inferences from Winsorized data are not substantially different from the case where the upper and lower α-fractions of the data are not used in model fitting, for example, as in the case of using *trimmed least squares* instead of ordinary least squares. Very importantly for big data studies, Winsorization in each variable leaves the database architecture and structure of data tensors unaltered throughout, which leads to computational simplicity.

In this paper, several different probabilistic and Bayesian ML and AI methods are studied—each of which derive their probabilistic nature from different aspects of a neural network architecture. We primarily conduct thorough empirical studies on several datasets in this project. For each probabilistic ML technique, we conduct a four-fold study of each dataset. First, we use the data as it is, unaltered. Second, we introduce independent and identically distributed Cauchy random variables in each of the target observation, thus creating an extremely noisy target with potentially several outliers of different magnitudes. The feature set is left intact. Third, we introduce independent and identically distributed Cauchy random variables in each observation in each feature, but leave the target intact. Fourth, we introduce independent and identically distributed Cauchy random variables in each of the target observations as well as each observation of all the features. Thus, the four versions of each dataset represent a (*i*) *no noise* scenario, (ii) a *noise in target* scenario, (iii) a *noise in features* scenario, and (iv) a *noise in target and features* scenario. Then, we analyze the dataset versions both as they are and when Winsorization is used to eliminate outliers. The goal of this study is to understand how probabilistic outliers affect the results from black-box ML techniques and how Winsorization may be used to greatly ameliorate the problem.

The rest of this paper is organized as follows: In Section 2, we discuss the related methods to introduce perturbations and train in the presence of adversarial noise. In Section 3, we discuss the different Bayesian deep learning methods that are employed for a comparative analysis. Section 4 outlines the experimental setup and results from the experiments. We discuss our findings in Section 5 and Section 6.

## 2. Related Literature

In several applications in computer vision, natural language processing and machine perception, deep neural networks have achieved remarkable performance. However, in the presence of even small perturbations in the training samples, the performance deteriorates quickly [6,7,8]. This instability makes it imperative to study the robustness of deep learning methods.

For instance, robustness analysis in domains such as natural language processing (NLP) often focuses on introduction of adversarial examples [9] while training neural networks. These multi-scale adversarial perturbations range from character-level to sentence-level perturbations. Several studies [10,11] also compute forward gradients of input sequences to guide the search for modifications that would introduce perturbation. Studies [10,12,13,14,15] on which a word’s addition or omission adversely affects model performance have shed light on the importance of different words in recovering from the adversarial attacks. Attacks based on gradient computation and insertion of perturbation in embedding space are susceptible to vanishing or exploding gradient problems. While, there are several black-box methods for adversarial attacks, their reproducibility in the NLP domain is also known to be limited [9]. To overcome these adversarial attacks, either the adversarial examples are identified and separated from the training set or the model architecture is modified to accommodate for the additional difficulty in learning. Adversarial example detection relies on recognizing known words and treating perturbations as unknown words that are not used for training [16,17]. Model modification based methods, on the other hand, generally include the adversarial examples during training [18,19,20]. The way similarity and dissimilarity are computed between adversarial examples and perturbation-free samples can also shed light on how can we optimally distinguish between the two. Through clustering of the embeddings of input words, shared encoding among similar embeddings can be used to differentiate the noise from the words [21]. Studies on constraining the perturbation in input have helped in development of certifiable defenses [22,23,24]. This certification can also be incorporated as an objective to create an adaptive regularizer that enhances the robustness and stability of the model [25]. Several studies also focus on Interval Bound Propagation (IBP) that propagate some verifiable input-output properties [26,27]. There is also evidence suggesting that overfitting on training data in overparameterized regimes adversely affects the performance of neural networks in presence of adversarial perturbations. The performance gains owing to all the adversarial training frameworks can be achieved by early stopping as well [28].

Similar to the application in the NLP domain, perturbation-based robustness in deep learning also utilizes adversarial training [29]. For a loss function of the form l(x,y;W), E refers to the expectation value, where W is the weight vector parameterizing a neural network, the optimization problem can be stated as follows [30]:(1)$$W^{*}\in \arg\min_{W}E\left[\max_{\delta \in \Delta }l\left(x+\delta ,y;W\right)\right]$$
where the perturbation is norm-bounded as, Δ={δ:||δ||≤ϵ}. For a worst case perturbation problem, we want to find δ such that it maximizes the loss function, and we also want to find the W array that minimizes the empirical risk.

Apart from artificially induced perturbations as listed above, there are several forms of natural perturbations in real world data as well. While it is important to study the synthetically generated perturbations to understand neural networks better, these are limited in their applicability due to their norm-bound restrictions and dearth in real world datasets, especially when data distribution shifts as it does in scenarios such as changing weather conditions [30]. In computer vision as well, naturally occurring noise in the form of blurring effects and distorted lights are less likely to be fully accounted for using norm-bounded perturbations [31]. More recently, model-based robust optimization methods are being employed to incorporate a natural variation that may take the form G(x,δ) and may already be a part of the training examples in the form of blurring and distortion, for instance. The objective, in this case, will be of the following form [30]:(2)$$\min_{W}E\left[\max_{\delta \in \Delta }l\left(G\left(x,\delta \right),y;W\right)\right]$$

G(x,δ) is called the model of natural variation and can be non-linear in δ. The model can be of encoder–decoder form as well allowing for learning an encoding for *G*.

Similar to perturbations, outliers can potentially contaminate any data analysis and provide misleading results. A lot of the recent attention has been placed on removing the outliers by truncating or trimming them. While this omission may be useful in removing the influence of extreme values, it still leads to a loss of information. Alternatively, the dataset can be “clipped” or “Winsorized” by replacing the extreme values with more central samples. Winsorization aids in managing the adverse effects of outliers in the data by clipping the extreme values. There have been several studies into efficiency and bias comparisons of Winsorized mean estimators [32,33] and Winsorized regression [34] where the residuals are Winsorized. Due to its simplicity, Winsorization may hold more appeal as compared to other robust regression techniques. There are also studies that attribute instability caused by perturbations not just as a shortcoming of model deep learning frameworks but also an attribute of adversarial training examples and the data itself. Outliers and perturbations in data make it difficult for the model to learn as the model regards both the clean samples and perturbed samples as equally important for training at the start of the learning process [35,36]. Winsorization as a data treatment addresses this concern and focuses on eliminating the outliers before the training begins. This way the association between target and features is preserved for the model to learn.

While the quantiles and median of error taken from the observed target might be robust towards outliers and make it favorable to incorporate in the learning objective, these functions are not differentiable, making them an intractable choice for gradient based learning. Recent advances in the space of robust neural networks learning include the use of M-averaging functions over the mean in the empirical risk estimation [37]. As an approximation of quantiles, a differentiable parametric family of M-average functions can be used such that they satisfy certain differentiability constraints and can act as surrogate for quantiles of loss, independent of outliers.

Winsorization has been used on regression problems using deterministic neural networks [38,39]. Asymptotic properties of trimmed and Winsorized M and Z estimators have been investigated and trimmed M-estimators have been used for robust estimation in neural networks [40]. As opposed to trimming or Winsorization of residuals, we seek to understand Winsorization as a treatment method on the training dataset. The aim of our study is to understand the effect of Winsorization on perturbed input data that are used to train probabilistic neural networks and investigate if Winsorization can aid in producing stable prediction results in probabilistic neural networks.

## 3. Methodology

We now briefly discuss the different probabilistic machine learning methods we study in this paper.

### 3.1. Exact Gaussian Processes

One of the most prominent and relatively simpler technique to use in prediction and inference in the presence of unknown functional relations between variables is the *Gaussian Process* (GP) modeling approach. This is a Bayesian approach which models unknown functions as Gaussian stochastic processes, that is, the evaluation of the unknown function on any finite collection of points in the feature space is assigned a multivariate Gaussian prior, and a posterior prediction is obtained by coupling the observed data with this prior. GP modeling is a non-parameteric regression approach with uncertainty quantification. GP regression can be fully specified using a mean and a covariance function that can be used to define a Gaussian probability distribution from which the predictions are drawn. GP is adept at capturing non-linear relations between the feature set and the prediction function using a non-linear covariance function kernel. A function f:X→R is modeled as a Gaussian Process with mean function *m* and covariance function *k* and can be written as follows [41]:(3)f∼GP(m(x),k(x)),
if for any finite integer k≥1 and any collection of points $x_{1},x_{2},\ldots ,x_{k}\in X$, the *k*-dimensional random variable $\left(f\left(x_{1}\right),\ldots ,f\left(x_{k}\right)\right)$ has a multivariate Gaussian distribution with mean $\left(\mu \left(x_{1}\right),\ldots ,\mu \left(x_{k}\right)\right)$ and a covariance matrix Σ whose (i,j)-th element is $k(x_{i},x_{j})$.

If f are the function values for the training set and $f_{*}$ is the set of function values on the test set $X_{*}\subset X$, the joint distribution can be specified as follows:(4)$$\left[\begin{array}{c} f\\ f_{*} \end{array}\right]∼N\left(\left[\begin{array}{c} \mu \\ \mu_{*} \end{array}\right],\left[\begin{array}{c} \Sigma \;\Sigma_{*}\\ \Sigma_{*}^{T}\;\Sigma_{**} \end{array}\right]\right).$$

Here, $\Sigma_{*}$ represents the train-test set covariance while $\Sigma_{**}$ represents the test set covariance. The conditional distribution of $f_{*}$ given f is as follows:(5)$$f_{*}|f∼N\left(\mu_{*}+\Sigma_{*}^{T}\Sigma^{-1}\left(f-\mu \right),\Sigma_{**}-\Sigma_{*}^{T}\Sigma^{-1}\Sigma_{*}\right)$$

More often than not, the mean function *m* and covariance function *k* involve hyperparameters, θ, that are tuned by maximizing the logarithm of marginal likelihood. The marginal likelihood gives us the probability of observing the data samples given hyperparameter values and is of the following form:(6)$$log\;p\left(y|x,\theta \right)=-\frac{1}{2}log\left|\Sigma \right|-\frac{1}{2}{\left(y-\mu \right)}^{T}\Sigma^{-1}\left(y-\mu \right)-\frac{n}{2}log\left(2\pi \right).$$

Partial derivatives of Equation (Equation 6) give us the gradient estimate update rules for the hyperparameters of the mean and covariance functions whose values can be calculated using an iterative numerical optimization technique. The exact GP has been proven to be very successful in several empirical use cases. Depending on the different kernel functions, the definition and shape of similarity that is encoded through the kernel function can be changed. Exact GP, however, becomes intractable for extremely big datasets as the computation cost scales by $O(n^{3})$, while storage scales by $O(n^{2})$ as *n*, the number of training samples, increases [41]. Therefore, several approximation methods have been devised recently to improve the scalability of Gaussian processes.

### 3.2. Variational Gaussian Processes

Exact Gaussian processes models are capable of utilizing high-dimensional feature sets for the modeling response. However, with increasing sample sizes in the wake of the advent of big data, the computational burden involved in defining the kernel function may increase in a super-linear way and as a function of the sample size. Therefore, it may be of interest to explore a more sparse kernel function [42,43] that can be defined in a more efficient manner, such as in case of Variational Gaussian Processes (VGP). Variational inference [44,45,46] also improves the efficiency in approximating the posterior predictive function. In this paper, we use a variational Gaussian process approach that renders the output of a deterministic deep neural network (DNN) as probabilistic.

Based on the mean field variation inference theory and use of hidden variables to encode representations from the observational data to obtain the posterior conditional distribution [47], a practical method utilizing sparser covariance structure is proposed to obtain a variational Gaussian process framework for big data [45]. Let u represent a vector of function values at a subset of samples $Z={\left\{z_{i}\right\}}_{i=1}^{m}$ from x called inducing variables. These inducing variables can be utilized to create a model that is consistent with the application of stochastic variational inference [47].

Marginalizing the inducing variables in the work [48], the lower bound on logp(y|x) can be obtained as follows [45]:(7)$$L=log\;N\left(y|,0,K_{nm}K_{mm}^{-1}K_{mn}+\beta^{-1}I\right)-\frac{1}{2}\beta \;tr\left(\tilde{K}\right)$$
where β is the precision of the original probability distribution of response conditioned on function f. Inducing variables u perform the role of global variables in applying a stochastic variational inference to a Gaussian process model. These are used to further lower the bound on p(y|x). The updated lower bound becomes the following: (8)$$L^{\prime }=\sum_{i}^{n}\left\{logN\left(y_{i}|k_{i}^{T}K_{mm}^{-1}m,\beta^{-1}\right)-\frac{1}{2}\beta {\tilde{k}}_{i,i}-\frac{1}{2}tr\left(S\Lambda_{i}\right)\right\}-KL[q\left(u\right)||p\left(u\right))].$$

The partial derivatives of $L^{\prime }$ provide us with estimates for the kernel hyperparameters and the noise precision β. Furthermore, a stochastic gradient estimate can be performed to obtain the optimal values for the variational parameters. The factorization of $L^{\prime }$ enables performing stochastic gradient methods on q(u) and also the use of non-Gaussian likelihoods for inference.

### 3.3. Concrete Dropout

Unlike the conventional use of dropout [49] to improve generalization power by sampling neurons during training, we can derive an empirical predictive distribution by using the layer-wise dropout relaxation during the testing process [50,51] where the variance of the predictive distribution is generated by randomly dropping neurons at test time using the optimal dropout rate. The optimal dropout rate can be found either by a grid-search over dropout probabilities [50], which can be prohibitive when computational resources are constrained. In this work [50], dropout is used to obtain an approximation to a probabilistic deep Gaussian process [52].

Similar to an $L_{2}$ regularization objective for dropout, variational parameters vector $\theta_{i}$ for each layer *i* can be regularized to achieve a model that reduces the Kullback–Leibler (KL) divergence of the weight distribution with the true Bayesian posterior as follows:$$\begin{array}{cc} F_{GP-MC}\;\propto & KL\left[q\left(W|\theta \right)||P\left(W\right)\right]-E_{q(W|\theta )}\left[logP\left(D|W\right)\right]\\ \;\propto & \frac{l^{2}}{2\tau N}\sum_{i}^{L}(p_{i}\left|\right|\theta_{i}{\left|\right|}_{2}^{2}+\left|\right|m_{i}{\left|\right|}_{2}^{2})-\frac{1}{\tau N}\sum_{n}^{N}logp\left(y_{n}|x_{n},W\right) \end{array}$$
where $m_{i}$ refers to the bias terms in each hidden layer *i*, $p_{i}$ is the dropout probability, *l* is length-scale, and τ is the precision hyperparameter.

Alternatively, a more efficient way of learning $p_{i}$ instead of doing a grid search is by setting layer-wise dropout rates as trainable and by learning them via the standard backpropagation process along with other neural network parameters. This method is called concrete dropout [51]. Similar to the KL divergence term in the grid-search scenario [50], the KL divergence term for the dropout rate estimation by variational free energy optimization may also include the variational parameters, thus:(9)$$KL\left[q\left(W|\theta \right)||P\left(W\right)\right]\propto \frac{l^{2}\left(1-p\right)}{2}{\left||\theta |\right|}^{2}-KH\left(p\right)$$
where *K* is the dimension of weight vector for each layer and H(p) is the entropy of dropout probability, which is a Bernoulli random variable in this case:(10)H(p)=−plog(p)−(1−p)log(1−p).

Moreover, a concrete relaxation of dropout masks makes it possible to obtain the optimal dropout probability value for each layer by pathwise derivative estimation [51]. If u∼Unif(0,1) and *t* is a temperature value, then the concrete distribution random variable will be of the following form:(11)$$\tilde{z}=\mathrm{sigmoid}\left(\frac{1}{t}\left(log\;p-log\left(1-p\right)+log\;u-log\left(1-u\right)\right)\right)$$

Concrete dropout does not require a lot of additional compute as compared to a standard dropout implementation and is more efficient than a grid-search to find optimal dropout probability value.

### 3.4. Flipout Estimator

Historically, there have been several advances in the field of regularization to overcome overfitting [49,53,54,55,56]. Some of these methods include Gaussian perturbations [57] and DropConnect [58] as methodologies for perturbing weights for regularization. Interestingly, while these methods were originally formulated for regularizing artificial neural networks (ANNs), the resulting stochasticity in weights also allows for the quantification of uncertainty in weights to some extent. Bayesian neural networks have been created by perturbing the weights in different hidden layers and trained using variational inference after applying reparameterization trick as elucidated in [59] that makes use of *Bayes by backprop* possible [60]. For an unknown weight parameter $W_{ij}$ for $i^{th}$ layer and $j^{th}$ node, $W_{ij}$ is drawn from $N(\mu_{ij},log{\left(1+exp\left(\Sigma_{ij}\right)\right)}^{2})$, where $\theta_{ij}=\left(\mu_{ij},\Sigma_{ij}\right)$ are variational posterior parameters that are trained via standard backpropagation. This is often done through the introduction of a non-parametric noise distribution, ϵ∼N(0,1), that is then scaled and shifted as follows:(12)W=μ+log(1+exp(Σ))⊙ϵ.

Here, using log(1+exp(Σ)) ensures that this term remain positive and differentiable. Variational free energy [57,61,62,63,64] is minimized to estimate the parameters θ of the weight distribution such that the Kullback–Leibler (KL) divergence with the true weights posterior is minimized [57,60]. Similar to previous work on weight-based uncertainty [60], the objective function remains as follows:(13)$$F_{Flipout}=KL\left[q\left(W|\theta \right)||P\left(W\right)\right]-E_{q(W|\theta )}\left[logP\left(D|W\right)\right],$$
which intends to minimize the negative log likelihood based on the data and the complexity cost of fully encoding the functional relation as a very complex W. This cost not only optimizes for the best weights for predicting our target but also optimizes for the simplest weight representation that we can get. The cost wants to reduce the number of bits required to transmit weights to a receiver, and this is the complexity cost of the weights and the second component of the loss function wants to minimize the number of bits required to transmit the errors in the model. These additive costs are together called compression cost [60] or minimum description length [57]. Through this cost function, we can ensure that the weight distribution is not too complex and does not overfit. Another form of Equation (Equation 13) uses an approximate cost function for complexity cost that makes the computation more efficient. Introduction of the perturbation ensures that the gradient estimates of the cost are unbiased [60]. Monte Carlo samples of W are drawn from the variation posterior distribution can be used for calculating the approximate cost.

In a similar fashion, adding parametric noise to weights has also aided in efficient exploration of optimal agent policies in reinforcement learning [65,66,67]. Depending on the network size, Gaussian noise added to the weights may be either independently used or factorized as well [67]. This means that for *p* inputs to a layer and *q* outputs, adding independent Gaussian noise to each weight will yield (p+1)q noise variables, while using factorized Gaussian noise would lead to individual noise weights for the noise and for the outputs such that only p+q noise variables are employed for a hidden layer. However, having the same Gaussian perturbation for a mini-batch may lead to some correlation in gradients and lead to high variance in the gradient estimates [68].

The flipout gradient estimation technique [68] can be used to de-correlate the gradient estimates. Like all the previously proposed methods, flipout also has a base noise that affects the ▵W where the base noise is drawn from a unit Gaussian distribution. This is similar to the previous weight based uncertainty [60]. Any other method that uses weight perturbation for weight-based uncertainty can be used as the method to obtain the gradient estimates that can be further de-correlated using flipout. Let the gradient estimates obtained from any of the previous work be $▵\hat{W}$. $▵\hat{W}$ in a mini-batch are still correlated. In order to de-correlate, we make use of randomly drawn sign matrices. If $r_{n}$ and $s_{n}$ are two random vectors drawn from ±1 uniformly, the flipout perturbation can be given as follows:(14)$$▵W_{n}=▵\hat{W}⊙r_{n}s_{n}^{T}.$$

Here, the *n* subscript represents each training example in a mini-batch. These additional matrix multiplications with sign matrices have been shown to decorrelate the gradient estimates and lead to variance reduction. Therefore, the convergence to optimal variational parameter values happens earlier. However, having the additional matrix multiplication with the sign matrices also leads to higher computational cost which may be offset by parallelizing computations through evolution strategies, similar to the ones used in reinforcement learning [66,69,70]. Similar to [57], evidence lower bound (ELBO) is optimized to obtain estimates for θ as is done in Equation (Equation 13). Through empirical evidence, it has been shown that flipout is able to reduce the variance in stochastic gradient estimates [68]. However, it is also stated that as compared to vanilla implementation of *Bayes by backprop*, flipout is 60 times more computationally expensive [68].

### 3.5. Mixture Density Networks

A mixture density network [71] allows for the response distribution to be a mixture of distributions, essentially allowing for multiple distributions to weigh in with varied degrees such that the response is multimodal. The parameters of each of these distributions are estimated using a fully connected artificial neural network. For solving inverse problems that often involve one-to-many mappings or for estimating multimodal distributions, a mixture density network (MDN) proves to be an effective regime. The conditional probability of obtaining a response value by mixing *k* Gaussian components can be given as follows [72]:(15)$$p\left(y|x\right)=\sum_{k}\pi_{k}\left(x\right)N\left(y|\mu_{k}\left(x\right),\sigma_{k}^{2}\left(x\right)\right).$$

Here, for each Gaussian component *k*, the parameters $\pi_{k}$, $\mu_{k}$, and $\sigma_{k}^{2}$ are estimated via a neural network with *k* outputs. $\pi_{k}$ is the contribution of the $k^{th}$ mixing component, $\mu_{k}$, and $\sigma_{k}$ are the mean and standard deviation of the Gaussian distribution that decide the mixing component’s density distribution. In addition, since $\sigma_{k}^{2}$ depends on the input x, this is considered a heteroscedastic model. If W represents the weight vector that specifies the neural network, Equation (Equation 15) can be written as follows:(16)$$p\left(y|x\right)=\sum_{k}\pi_{k}\left(x,W\right)N\left(y|\mu_{k}\left(x,W\right),\sigma_{k}^{2}\left(x,W\right)\right).$$

The weights W can be estimated by minimizing the negative logarithm of likelihood which will be our loss function that we will optimize for. The loss function is of the following form [72]:(17)$$F_{MDN}=-\sum_{i}log\left\{\sum_{k}\pi_{k}\left(x_{i},W\right)N\left(y_{i}|\mu_{k}\left(x_{i},W\right),\sigma_{k}^{2}\left(x_{i},W\right)\right)\right\}.$$

Minimizing Equation (Equation 17) gives us the conditional density function of the response. Varying the number of mixing components *k* may also be an additional tuning hyperparameter as it decides the number of modalities in the dataset.

### 3.6. Winsorization

In Winsorization, the extreme values are replaced with more centrally located representative values. The samples within the first α percentile are replaced with the $\alpha^{th}$ percentile sample, and the samples beyond the 1−α percentile are replaced with ${\left(1-\alpha \right)}^{th}$ percentile value. This is similar to clipping that is used in gradient vanishing/explosion problems. By bounding the dataset to exclude the outliers, the adverse effects of perturbations on the training are attenuated.

For a given input random variable *X*, the values in *n* training examples can be written in the form of ordered statistics as below:(18)$$X_{\left(1\right)}\leq X_{\left(2\right)}\leq \ldots \leq X_{\left(\alpha n\right)}\ldots \leq X_{\left((1-\alpha )n\right)}\ldots \leq X_{\left(n-1\right)}\leq X_{\left(n\right)}.$$

After Winsorization, the sequence will be modified as follows to include more of centrally located samples:(19)$$X_{\left(\alpha \right)}\leq \ldots \leq X_{\left(\alpha \right)}\leq \ldots \leq X_{\left(1-\alpha \right)}\leq \ldots \leq X_{\left(1-\alpha \right)}.$$

As a methodology for limiting the effect of outliers, Winsorization can be applied on the training dataset for more robust learning in probabilistic neural networks. To test for robustness, we induce artificial perturbations in the training and validation set in features and targets by adding standard Cauchy noise. The model performance is evaluated on an untouched test set with no perturbation and a test set with perturbations. Winsorizing the training data ensures that the model is able to learn on more central values in feature and response space.

In the presence of outliers, we have the following:(20)$$sup\;|Y-\hat{Y}|\geq sup\;|Y-{\hat{Y}}_{W}|$$
where ${\hat{Y}}_{W}=f\left(X_{W},Y\right)$ or ${\hat{Y}}_{W}=f\left(X,Y_{W}\right)$ or ${\hat{Y}}_{W}=f\left(X_{W},Y_{W}\right)$. Here, *f* is the learned neural network based estimator for the response *Y* based on the input *X*. In the training dataset, if either the input space or the response space are clipped, it affects the learning of the neural network model.

## 4. Results

The performance of the models are evaluated on five datasets. Apart from comparing the performance of different probabilistic modeling methods, the experiments also provide insights into the use of Winsorization in the training stage of neural network modeling.

### 4.1. Datasets

#### 4.1.1. Precision Agriculture Case Study: Crop Yield Prediction in the US Midwest

Historically, process-based biophysical models and classical statistical models have been employed for crop yield prediction. Process-based models [73,74,75,76,77] study physiological and physical processes to simulate crop yield. Often, simpler statistical models [75,78,79] are used owing to their straightforward reporting of goodness of fit metrics. Many of these models may be constrained by their strict and often unrealistic assumptions to control multi-collinearity and spatio-temporal dependence [80]. Additionally, process-based and simple statistical models often miss or exclude non-linear terms, which may prevent them for being useful for yield predictions under extreme climatic conditions. Machine learning methods present the opportunity to model agricultural data using more complex architectures, using fewer assumptions, and on larger datasets. Artificial neural networks [81,82,83,84,85,86,87,88,89], linear regression [87,88,90,91,92,93], tree-based models [87,92,93,94,95,96,97,98], and support vector machines [98,99,100] are some of the most used machine learning algorithms [101,102] for crop yield modeling. In particular, ANNs have been used for tasks such as species recognition, weed detection, or crop quality assessment in [81,82,83,84,85,86,87,88,89] and elsewhere, using a variety of complex features including satellite data.

In applications of statistical machine learning modeling to climate sciences and precision agriculture, it is important to incorporate spatio-temporal dependencies between multiple samples. Spatial and spatio-temporal statistical models may be used to capture such dependencies explicitly but are very sensitive to the stringent assumptions made for such models, and the computations do not scale with data size in these cases. Ignoring possible non-linear, complex functional relations might lead to considerable over-estimation of the uncertainty in prediction and a loss of statistical power for feature selection and risk bounds, which has severe consequences for downstream industries such as that of crop insurance. More alarmingly, under-estimating uncertainty may lead to misleadingly narrow uncertainty bounds that may be centered at an inaccurate and biased estimate. In view of these issues, this work focuses on using easily available within-season meteorological variables for rapid and efficient usability and ensures that ANNs capturing non-linear functional components are used to tackle all of the above listed concerns. Several probabilistic modeling techniques are explored in this section.

Studying the effect of variations in weather on human crop yield is important in mitigating production and economic losses in agrarian economies. This observed effect is conspicuous and well studied in [103,104]. Extreme weather events in the US midwest affect crop yields, food price hikes, and can lead to production losses—the effect being as high as 75% for some Minnesota counties [103]. Formulating data-driven methods is imperative for more efficient planning in precision agriculture and for public policy. In view of this, our experiments are performed on a climate dataset comprising of county-level end-of-year crop yields in the Midwest USA and daily meteorological variable readings. The data include daily maximum temperature readings, minimum temperature readings, and average precipitation readings. These help us in gauging how the end-of-year yield changes in a county depending on the climatological features and location. The feature set includes daily maximum temperature readings (365 features), daily minimum temperature readings (365 features), daily average precipitation readings (365 features), longitude and latitude (2 features), and cosine and sine transformation of these location coordinates (4 features). The meteorological data are from the PSL public dataset [105], and the corn crop yield data are from the USDA public dataset [106]. The dataset is limited to Minnesota and Illinois for clarity of presentation.

#### 4.1.2. California Housing Data

The California housing data [107] are a well-known public dataset. The response or target variable is the median house value for different California districts collected in the 1990 U.S Census. There are eight features including median income of the block, median house age, average number of rooms, spatial coordinates, and population in the block, where a block is the smallest geographical unit in the Census. Algorithmic bias can unduly affect the model performance of predictive models. Especially, there may be confounding variables that lead to the model favoring certain sections of the population, and it is imperative to understand the extent of influence of this bias. Using probabilistic deep learning models ensures that the some transparency is lent to the black-box model in making predictions.

#### 4.1.3. Data on Forest Fires in Portugal

The dataset on forest fires [108] contain information on the area burned by forest fires in Portugal using meteorological data. The response variable is a scaled burned area variable. The features include spatial coordinates, temperature, humidity, rain, wind, information from the Fire Weather Index (FWI) system, and time-based variables. Many times, the burned area is an estimate based on surveying methods. It is also possible that the feature values such as humidity and temperature are often measured in a metropolitan location instead of at the center of the forest area. These may make our estimates more biased due to lack of accurate feature information. Skewed observed target values and noisy data can have critical implications on the human lives and wildlife that are affected by wild fires each year. Obtaining uncertainty estimates will allow policy makers to analyze all probable prediction scenarios.

#### 4.1.4. Mauna Loa CO${}_{2}$ Data

This dataset consists of time series atmospheric CO${}_{2}$ concentration variations data as collected by NOAA at Mauna Loa, Hawaii [109]. The data consists of monthly atmospheric CO${}_{2}$ concentration readings from the year 1958 through August 2021. Climate change has severe implications on the green house gas concentration in the environment. Bayesian deep learning methods enable the revision of our known knowledge and provide better forecasts for climate-change-driven planning.

#### 4.1.5. Data on Genomics of Drug Sensitivity in Cancer

Genomics of Drug Sensitivity in Cancer (GDSC) [110,111] study involves studying the sensitivity of cancer cells to anti-cancer drug as affected by the gene expressions of different patients. The GDSC database from which the data is derived curates over thousands of genetically characterized human cancer cells and information about biomarkers.

Each cancer patient responds differently to anti-cancer drug treatment. The objective is to explore therapeutic biomarkers that may be used in identifying patients that are more likely to respond to anti-cancer drug treatment. In precision oncology, we are interested in providing the correct drug treatment to the right cancer patients based on these biomarkers of drug sensitivity.

In the dataset, the response variable is the sensitivity to YM155 (Sepantronium bromide) as the natural logarithm of the fitted IC50 and the feature information consists of expression of 238 genes. The dataset samples are drawn from 4 cancer types and 148 cell lines. Precision in medicine can improve human mortality and uncertainty-guided model inference can lead to more reliable research outcomes in healthcare.

### 4.2. Architecture

We use a fully-connected deterministic neural network as the baseline model. It has 14 layers and ReLU activation for the first 13 layers. The number of hidden layers are 52, 250, 300, 450, 202, 452, 50, 300, 200, 52, 400, 350, 450, and 450 and 1 for the output layer. This network is chosen via Bayesian optimization. This involves using expected improvement as a surrogate acquisition function. This function is the objective function for neural network hyperparameter optimization. Over several trials using the surrogate function with different hyperparameter sets, we choose the model with optimal hyperparameter set, and this chosen hyperparameter set can be expected to accurately correspond with the maximized original objective function. The expected improvement for ${\left(x_{i},y_{i}\right)}_{i=1}^{n}$ samples and ${\left[a\right]}^{+}=max\left(0,a\right)$ is given by:$$\mathrm{Expected}\;{\mathrm{Improvement}}_{n}\left(x\right)=E_{n}\left[{\left[f\left(x\right)-f_{n}^{*}\right]}^{+}\right].$$

More information about Bayesian hyperparameter tuning can be found in [112].

*Concrete Dropout*. One of the Bayesian neural networks we experiment with implements the concrete dropout methodology on all layers. The dropout rate is learned during the training stage, and the contribution of different hidden neurons is varied by dropping neurons during the testing stage. Mean and variance of prediction distribution are the outputs of the neural network. The ELBO function is modified to include the concrete relaxation for dropout rate. This allows for a more efficient search for optimal dropout rate and probabilistic model training as opposed to Monte Carlo dropout learning based on grid search.

*Variational Gaussian processes (VGP)*. Another implementation uses a variational Gaussian process to estimate the target. The first 14 layers learn weights in a deterministic manner. The VGP learns the posterior predictive distribution of the target and receives its input from the sequence of fourteen fully connected layers. Instead of full covariance we use a sparser representation using inducing variables. An RBF kernel is used with amplitude and lengthscale hyperparameters. The method [45] is different from previous variational GP implementations in that the inducing variables and the kernel hyperparameters are jointly optimized using gradient descent-based updates.

*Flipout Gradient Estimator*. In the flipout implementation, gradients in all 14 layers are estimated using the flipout method [68]. The KL divergence of the bias and kernel surrogate posteriors and priors are minimized during training. Flipout enables variance reduction in the gradient estimates by minimizing the KL divergence of the weights’ variational posterior with the prior on the weights. Apart from minimizing the complexity cost, we are interested in improving the expected logarithm of the likelihood of observing the data when the weights are sampled from the variational posterior distribution. The objective function comprises of both these terms and is called the compression cost. The gradient estimates are taken with respect to an approximate version of the variational free energy cost where the weight values are sampled using Monte Carlo sampling from the variational posterior.

*Mixture Density Networks*. Here, the mixture components are estimated via the fully connected neural network. The estimated parameters are used to estimate the response using Gaussian mixture models. The mixing coefficients as well as the mean and variance of the Gaussian components’ density are estimated using the neural network. The gradient estimates of the negative log likelihood function are used to update the parameter values.

#### Evaluation Metrics

For training purposes, the datasets are split into training, validation, and test sets, which comprise of 70%, 10%, and 20% of the data. Since all the experiments essentially relate to regression problems, we use mean squared error as our metric of comparison. We also report the median absolute error which provides insight into the model performance in presence of outliers. The total time for training one instance of the model is reported. We also report the coefficient of determination and the total number of parameters in the network architecture. Apart from the methodology-based change in number of model parameters, the number of parameters also vary depending on the number of input features for the different datasets. The effects of adding artificial perturbations to the dataset and using Winsorization to mitigate the effect of the noise is also studied. The evaluation metrics are reported for different sites for noise and for different degrees of Winsorization. The effect is tested on a test set that is free from noise (original, untouched test set) and a test set that has standard Cauchy noise at different sites, the same way as the training/validation data sets (contaminated test set). Relative efficiencies are also evaluated to compare the Winsorized results relative to the non-Winsorized prediction results. We compare the efficiencies for different sites of contamination using standard Cauchy noise and at different degrees of Winsorzation.

### 4.3. Results

#### 4.3.1. Crop Yield Estimation

Table 1 outlines the results of crop yield prediction based on conventional deterministic learning methods with the exception of a shallow concrete dropout model. First, we implement a LASSO [113] regression approach, which imitates the generalized linear regression models that are popularly used in many of the related domain science outlets. The regularization coefficient at 0.0781 is learned adaptively through fivefold cross validation. Based on the LASSO regularization and resampling inference on deep, semi-parametric modeling (not shown here), we can infer a strong negative correlation between agricultural yields and harvest-period precipitation. Next, we use a random forest model with 100 trees and a max depth of 100 per tree. Next, a support vector regression with a radial-basis kernel and C=10 and a margin of error ϵ=0.1 is learned. To enable comparison with [79], we also fit a support vector model of 8th degree, with C=1.0. A shallow ANN model using three layers, with concrete dropout, is also fitted to the data. The deterministic deep ANN architecture is then used, which results in the best MSE and $R^{2}$ values on test data.

Table 2 reports the probabilistic frameworks that were tested on crop yield estimation problem. The results include variations of flipout based and MDN based networks. Apart from experiments with flipout gradient estimation on all layers, we also try the gradient estimation technique on specific layers in neural network. The flipout implementation on “early 5 layers” focuses on applying the estimation technique only on the first 5 layers of the model. Similarly, “mid 5 layers” focuses on applying flipout on the middle 5 layers while “final 5 layers” focuses on applying the method to the final 5 layers closer to the output. Flipout estimation for all layers not only leads to more number of floating point operations per second but also fails to estimate optimal parameters that may improve predictive performance. By applying flipout on subset of layers, we allow for the network to estimate the output in a more efficient manner and still retain the probabilistic behavior of the model (and the unbiased gradient estimates in the presence of weight perturbations). Figure 1 displays the prediction results for the flipout experiments. The colder counties in Minnesota have lower yield while warmer counties in Illinois have higher yields. For these relatively colder counties and relatively warmer counties, the model is still unable to accurately predict crop yield. This is visible in the lower left-hand-side tail of prediction in Minnesota result sub-plots and the upper right-hand-side tail of predictions in Illinois sub-plots. Similarly, variational GP is unable to achieve the same level of performance as exact GP. In the MDN, we vary the number of mixing components. As we increase the number of components, we add enough complexity to model multiple geographical regions using different mixing densities such that the multiple modalities of the data are sufficiently captured. This helps in improving the performance in terms of the test mean squared error and coefficient of determination. Figure 2 shows us the different predictions results on a geographical map. Flipout and VGP-based predictions are unable to cover the full range of the observed target. In Figure 3 as well, the prediction results can be noted by state. Bayesian neural network methods have lower predictive performance on Minnesota as opposed to Illinois. Epistemic uncertainty is higher for the concrete dropout and exact GP and lower for the mixture density network variants. While our inadequate knowledge about the best dataset and predictive model for supervised learning is displayed in the epistemic uncertainty, more discussion sources of uncertainties can be found in Appendix C.

**Winsorization Results**.

Experiments to test the effectiveness of Winsorization in removing Cauchy noise from the data are performed. In four different variations, we add standard Cauchy noise to *(a)* target, *(b)* features, *(c)* target and features, *(d)* neither target nor features in the training and the validation set. We experiment with two scenarios—when the test set contains no Cauchy noise in none of the four variations listed before and when the test set contains Cauchy noise according to the first three variations listed before. We compute the mean squared error, median absolute error, mean absolute error, and the coefficient of determination as our evaluation metrics. Table 3 shows the values of the metric before Winsorization as MSE, Median AE, MAE, and R${}^{2}$, respectively. The metric values after Winsorization are shown as MSE${}_{W}$, Median AE${}_{W}$, MAE${}_{W}$, and R${}_{W}^{2}$. For each noise site variation, an optimal Winsorization limit that minimizes the mean MSE${}_{W}$ is chosen to be displayed. The untouched test set is more similar to the central data distribution in the training set. With the exception of exact GP, the model performance on the test set unconditionally improves when the training and validation sets are Winsorized. In the contaminated test set, there is no apparent trend in model performance as Winsorization is applied. This is consistent with the fact that the model is training to learn the noise as well, increasing the coefficient of determination and reducing MSE values in the contaminated case as compared to the untouched test set case.

Figure 4 displays the Test MSE${}_{W}$ values on the untouched test set after Winsorizing the training and validation data set. The figure showcases results when perturbation is added to separate noise sites. Figure 4e,f show the results from Figure 4c,d only for the test MSE range 0 to 10. It is evident that, in general, the presence of noise using some degree of Winsorization in the training and validation set improves the model performance as opposed to when Winsorization is not used. The benefits of Winsorization are not clear when there is no artificial perturbation in the data. The marginal improvement in Figure 4a could be the result of the removal of naturally occurring outliers. The benefits of Winsorization are more apparent in Figure 4b–d. In addition, while exact GP was the best performing model before the addition of noise, the performance degrades after the addition of noise, especially when added to the feature set. Mixture Density networks perform relatively better for all the noisy cases. Probabilistic neural network models are able to overcome the effects of perturbation in features at relatively lower Winsorization levels but require a higher degree of winsorization to reach the optimal Winsorization limits for achieving the best test MSE.

#### 4.3.2. California Housing Data

For the California house price dataset, Table 4 shows the results for different probabilistic models. Similar to the previous case study, the variational GP and flipout gradient estimations in the presence of weight perturbation are unable to perform well. The mixture density network with two mixing components is able to perform well, but exact GP provides the best performance in the absence of any noise.

Figure 5 shows the Test MSE${}_{W}$ on the untouched test data as the degree of Winsorization on the training and validation sets are varied. Similar to the results obtained in the previous case study, there is no apparent certain improvement with varying Winsorization limits in test MSE${}_{W}$ when there is no noise in the data. When Winsorization limit is increased from 0 to 1 percentile and 5 percentile, we notice improvement in the performance for the probabilistic neural network models. The mixture density network is able to perform marginally better than exact GP when the Winsorization limit is 5 percentile. Beyond the 5 percentile mark, we notice that mixture density network with 2 mixing components are able to perform better than exact GP when the Winsorization limit is 15 percentile and 20 percentile. As noise is added to the features, the model performance for concrete dropout and mixture density network (with four components) improves from the pre-Winsorization case. It is visible in Figure 5b,d,f, that as the degree of Winsorization increases in the training/validation sets, the model performance of most probabilistic neural network models increases. In Figure 5c,e as the Winsorization limit increases from 0 to 1 percentile, we see the most model performance improvement. While the model performance continues to improve with the increasing Winsorization limit for the concrete dropout Bayesian neural network, we are unable to notice the same for other models. We notice that noise adversely affects exact GP’s performance.

In Table 5, the model evaluation metrics are provided on the untouched and contaminated test sets before and after Winsorization is applied. The optimal Winsorization results are shown. We notice similar trends as the previous cases study results. The optimal Winsorization limit is usually well below the 25 percentile threshold for the case when we introduce noise in the features before model training. For the untouched test set case, the model performance improves for all models including exact GP.

#### 4.3.3. Mauna Loa CO${}_{2}$ Data

The CO${}_{2}$ concentration in the atmosphere increases as a function of time, measured in months and years. A naive fitting on the CO${}_{2}$ concentration as response will lead to sub-optimal training. The model performance results and the Winsorization results on unadjusted responses are given in Appendix B. We therefore adjust for the long-term linear trend and monthly seasonal trend in CO${}_{2}$ concentration. Therefore the response, CO${}_{2}$ concentration, can be decomposed as follows:(21)$$Y_{i,k}=\mu_{i}+\mu_{i,k}+f\left(X_{i,k}^{\prime }\right)$$
where $\mu_{i}$ is the linear long term trend for the $i^{th}$ observation. This component can be fit as a linear regression model on $X_{i}$. $\mu_{i,k}$ is the seasonal mean for the $i^{th}$ observation and $f(X_{i}^{\prime })$ is the non-parametric fitting on the residuals using a probabilistic neural network. Here, $X_{i,k}^{\prime }=Y_{i,k}-\mu_{i}-\mu_{i,k}$. In addition, the linear long-term trend can be fit using the original features as follows:(22)$$\mu_{i}=X_{i}\beta$$
where the $\hat{\beta }$ estimator can be obtained by ordinary least squared for this dataset. Using this semi-parameteric model allows for the non-parameteric component of the model to focus on fitting to the more complex relationships without having to accommodate for the linear long term trend and the monthly seasonal trend. Table 6 shows the performance results for various probabilistic methods on the Mauna CO${}_{2}$ concentration dataset. Flipout is unable to converge to a representative posterior predictive distribution. Variational GP performs slightly better. Concrete dropout, mixture density networks, and Gaussian processes perform the best where MDN has slightly faster training time.

Figure 6 show the Test MSE${}_{W}$ on the untouched test set as the Winsorization limits on the training and validation set are increased from 0 to 25 percentile. In the noise-free case, as the degree of Winsorization is increased, test set MSE remains the same for the exact GP and changes marginally for MDN with 2 components and concrete dropout. For the case when noise is introduced in features, test MSE increases with the increasing degree of Winsorization. This may be due the fact that the feature set dimension is one and adding perturbations greater than feature values to all training examples adversely impairs the models from learning. When the noise is added at least to the target, there is a decline in untouched test set MSE. In the case, where there is noise only in target, the MDN performance improves drastically. The exact GP remains unchanged while concrete dropout improves marginally. For the case where there is noise in both target and feature, the exact GP remain the same throughout while the model performance for other models only improves to a certain point, even at a higher degree of Winsorization.

Table 7 displays the change in model performance. Similar to previous data sets, the model performance improves with Winsorization for the untouched test set while it does not for the contaminated test set. For the untouched test set case, while there is a Winsorization limit for which most methods see an improvement in performance, the model performance improves only slightly for the concrete dropout case.The optimal limit remains high for all noise cases on the untouched test set.

### 4.4. Forest Fires Data

Table 8 compares the model performance without any noise in the dataset and without Winsorization in the training and validation set. The forest fires dataset is a challenging dataset for all methods. None of the methods are able to achieve reasonable model performance on this dataset, suggesting that there is a need for exploring more methodologies and architectures in further work. Among the methods that were tested, similar trends as before were noticed. Flipout is the most expensive and takes more time. Exact GP and MDN achieve the best model performance and run time. VGP performs slightly worse and takes more time to train than MDN and Concrete dropout.

Figure 7 shows the change the model performance on the forest fires dataset when the training and validation datasets are Winsorized. In the noise-free case, there is no affect on the exact GP performance as the degree of Winsorization is increased. MDN performance remains worse than Concrete dropout and exact GP despite the changing Winsorization limits. For other cases, where noise is introduced, we notice that some degree of Winsorization helps improve model performance as opposed to the no Winsorization case, especially when there is noise in the response variable. In the cases where noise is introduced, Concrete dropout and exact GP are able to achieve the lowest Test MSE${}_{W}$.

Table 9 showcases similar trends as the previous datasets. The model performance improves with Winsorization on the untouched test set and there is no definite improvement for the contaminated test set. The optimal Winsorization limit for noise in the features case is low, similar to all datasets.

#### GDSC Data

As can be seen in Table 10, the GDSC dataset also proves to be a difficult dataset for the probabilistic models studied in this work. Among the models that were tested, VGP is able to achieve the best model performance followed by MDN with two mixing components.

Figure 8 shows the model performance on the untouched test set, and the Winsorization limit is varied on the training and validation set. Apart from the noise in the target case, all cases show no definite improvement in model performance as the Winsorization limit is increased. For the noise in target case, the model performance improves as the degree of Winsorization is increased. However, in all cases, the exact GP performance is not affected by the varying Winsorization limits.

Table 11 shows us the model performance on the untouched test set and the contaminated test set for the optimal degree of Winsorization for different noise sites. For the untouched test set case, we notice marginal improvement in model performance for all probabilistic neural networks. For the contaminated test set case, we notice that the model performance improves or remains the same for concrete dropout and exact GP. For the mixture density networks, performance improvements are noticed for only a subset of cases.

## 5. Discussion

Winsorization of the noise site in the presence of noise is able to aid in mitigating the adverse effects of outliers on the model performance for probabilistic neural networks. This can also be elucidated by measuring the relative efficiency (RE) of the Winsorized model performance with the pre-Winsorized model performance. RE can act as a metric that concisely conveys the impact of Winsorization on the test MSE. Figure 9 shows the logarithm of RE values, where RE is computed as follows:(23)$${\mathrm{RE}}_{W}\;(\mathrm{Relative}\;\mathrm{Efficiency})=\frac{\mathrm{Test}\;\mathrm{MSE}}{\mathrm{Test}\;{\mathrm{MSE}}_{W}}$$

An RE value of one represents the same level of MSE while an RE value greater than one signifies model improvement with Winsorization. In most cases, Winsorization leads to an improvement in model performance. In the crop yield dataset, where we suspect that there are natural variations in the data apart from artificial noise that may be interfering with the models’ ability to learn effectively, the RE values are always greater than one for all the noisy cases, barring exact GP. Exact GP is able to retain the same level of performance as pre-Winsorised learning when the noise is introduced in features but has improved performance when the noise is introduced in target. In other datasets as well, exact GP is not always able to see drastic model performance improvement. For more difficult datasets such as crop yield, GDSC, and forest fires, exact GP is not always the best performing in the presence of noise. For all datasets, the model performance is able to improve with Winsorization, especially when the noise site is target, especially at higher degrees of Winsorization. We see similar results when the noise site is target and features. The results on GDSC dataset are not indicative of definite improvement with the Winsorization of the training and validation data. On the other hand, the crop yield dataset shows the most improvement for all noise sites. This might be due to the fact that the architecture of the underlying fully connected neural network is optimized for the crop yield dataset using hyperparameter search by Bayesian optimization. Therefore, it may be meaningful in future work to investigate the effects of changing the architecture to be more suited for particular datasets.

We can also compare the Winsorized results in the noisy cases with the noise free, non-Winsorized results. The case where there is contamination only in feature set allows for comparison of MSE on target variable in contaminated data with noise-free data. In Figure 10, the black dashed line is the unattainable gold standard of achieving the same results as the noise-free training in the presence of noise. Our objective is to come as close to it as possible. In the results shown here, we make the comparison for crop yield dataset, which has optimized architecture design and the most stable of all data use cases presented here. In Figure 10a, it is shown that lower degree of Winsorization in the training and validation datasets helps improve performance over noise free data for the probabilistic neural networks. As the degree of Winsorization increases, the loss of information adversely affects the exact GP performance. Figure 10b displays the relative efficiencies on the untouched dataset for the noise in features case as well. For all the neural networks, the relative efficiency increases as we Winsorize the training and validation datasets. For exact GP, the relative efficiency does not change as Winsorization limit is increased. However, the model performance as indicated by relative efficiency does not remain at the same level as in the noise-free, non-Winsorized scenario. The neural networks are able to recover similar level of performance as the noise-free, non-Winsorized case. However, a higher degree of Winsorization lead to a loss of information that adversely affects the model performance for all probabilistic neural networks. For the untouched test set, the RE for feature noise site as compared to the RE when noise site is target (not shown here) indicate that for different datasets, Winsorization helps the most when the noise is in the feature itself. The RE results on feature noise site (not shown here) indicate that a lower degree of Winsorization (up to 10 percentile in our experiment results) aids in recovering the original model performance.

Figure 11 shows the average model performance over all noise sites (feature, target, and both feature and target) for different Winsorization limits on the untouched test set. The median absolute error and mean squared error are scaled by the maximum value of the respective metrics for each dataset. The mean and variance of the metrics are mentioned in the legend for reference. We notice a general trend of model improvement up to a certain limit. We also notice that GDSC is unable to achieve model performance improvement. For the more challenging datasets, it may be material to optimize the architecture to obtain more stable models for further experimentation.

Similar to comparison of model performance in terms of the mean squared error, the uncertainty of different probabilistic models can also be compared. Figure 12 shows how uncertainty in prediction changes as the degree of Winsorization is increased. While exact GP uncertainty estimates are mildly affected by change in Winsorization limits, there are discernible slight changes in uncertainty for other methods. As noise is added to target, the uncertainty estimates increase for a subset of datasets. For the datasets where this is visible in the pre-Winsorization uncertainty estimate values, with Winsorization, uncertainty estimates drop to a lower level as the degree of Winsorization is increased. Adding noise in features does not heavily influence the uncertainty estimates. Even in cases where there is noise in target and features, we can see a decrease in uncertainty estimates as the training and validation data are Winsorized. However, the erratic behavior of uncertainty estimates for mixture density networks requires further investigation into the source of volatility.

## 6. Conclusions

We compare different probabilistic neural networks in terms of model performance and time taken for training. Among the different methods that we employ, VGP based neural network, flipout, and concrete dropout solely rely on variational free energy for learning the variational posterior distribution (optimizing variational hyperparameters). MDN uses the negative log likelihood for estimating parameters that define response distribution. Exact GP also depends on optimizing for the marginal log likelihood to estimate the hyperparameters. In terms of the model performance, we see varied results for the different methods on different datasets. In general, when the flipout gradient estimation is used for all layers, the model performance and training time are adversely affected. The additional matrix computations in flipout make it more expensive. Recent literature also sheds light on the heave-tailed distribution of deep Gaussian processes [114,115]. Further experimentation on methods such as concrete dropout that try to provide approximation of deep Gaussian processes can be conducted to understand the properties of the predictive distribution arising from the implementation of such methods.

We experimented with the use of Winsorization to make probabilistic deep learning models more robust against outliers. Over several data sets, we obtained several model performance results for different noise sites and degrees of Winsorization. Through the results that we observed on the untouched test dataset, we are able to observe the effects of introducing noise and Winsorization on the training and validation dataset. Using an untouched test set enables an easy comparison with the original performance of the models on noise-free, pre-Winsorized datasets. We also obtained model performance results on noisy test set data. This helps us in exploring more realistic use cases where it is known that the whole data set is contaminated from noise or perturbations. For our case study on the crop yield dataset, it is shown that in the presence of noise in the features, Winsorization helps the models in recovering model performance, both when tested on untouched test set, and contaminated test set. Noise in the dataset drastically degrades the Exact GP performance. It further worsens by the loss of information as the degree of Winsorization increases.

We notice that for several experiments, Winsorization produces unfavorable results due to a loss of information. This much has been noticed for linear regression problems [116] as well. It has been shown that the Winsorization of the features and response increases the Mean Squared Error of the regression and also increases the variance in the estimates of the coefficients. As we have seen in the noise-free scenario in our experiments, this is not always the case for the Bayesian neural networks. Through the results shown in contaminated test data case, we see that the models learn the noise along with other signals in the data, as is evident from MSE values and coefficient of determination on the pre-Winsorized data. Winsorization is unable to improve the performance as the i.i.d. Cauchy perturbations degrade the quality of training that happens for the neural networks. On the other hand, the untouched test set does not have the artificial perturbations in the hold-out set, enabling us a direct comparison of the evaluation metric results with the noise-free dataset results. For most cases, Winsorization in the training and validation dataset clearly improves the ability of neural networks to learn the more centrally located values in the dataset. The performance change on the untouched test set suggests that a lower degree of Winsorization on the dataset might be beneficial in training. Meanwhile, the evaluation results on the contaminated test set may point towards the non-existence of universally stable neural networks in today’s deep learning frameworks that can withstand any perturbations [36]. These results also follow the deeper cause of the instability—the functional relation between the input and target are often based on correlations existing in the observed, real-world data. These functional relations are often brittle and disrupted by the perturbations. From the model’s perspective, the noisy and non-noisy samples in the data are potentially equally important for learning effectively at the start of the training process. These perturbations are learned by models, and often several models that learn independently on the same data learn the adversarial perturbations the same way. This perturbation learning can be transferred from one model to another, where each is trained independently. This is called adversarial transferability [35]. Similar to our results, this is proven on image classification problems where the evaluation metric results on the untouched test set yields improved accuracy. This suggests that robustness as a challenge is not only tied to the training of the models but is also a property of the dataset, making treatments such as Winsorization amenable to more effective learning.

Despite the vast literature focusing on the robustness of neural networks against adversarial attacks, producing a robust framework is still a daunting problem. It is therefore meaningful to study the effects of Winsorization on the instability that perturbations cause in neural network training. Instead of introducing perturbations to all samples in the data, as we did in our experiments, studying the effects of the sparser addition of perturbations to certain samples or certain features may be closer to the experimental setups in the current studies. In our study, we saw that Winsorization on the feature set in the presence of noise aided in obtaining good evaluation results. Choosing a Winsorization limit more adaptively for individual features may enable more effective learning as certain features may be more prone to instability arising from perturbations than the other features. A comparative analysis of the effects of Winsorization with other methodologies to deal with noise and outliers would also provide insights into the relative efficacy of these methods [35,117].

It is also a well-known result that training for several epochs does not adversely affect the generalization capabilities of neural networks, especially in overparameterized regimes. The same has been proven to not be true when training robust networks that deal against adversarial attacks. In fact, it has been shown that early stopping during the learning process can be more beneficial than several of the adversarial training algorithms that have been proposed to deal with adversarial examples. It would be interesting to explore how tuning the number of epochs for early stopping in our experimental setup would change the results.

## Figures and Tables

**Figure 1 entropy-23-01546-f001:**
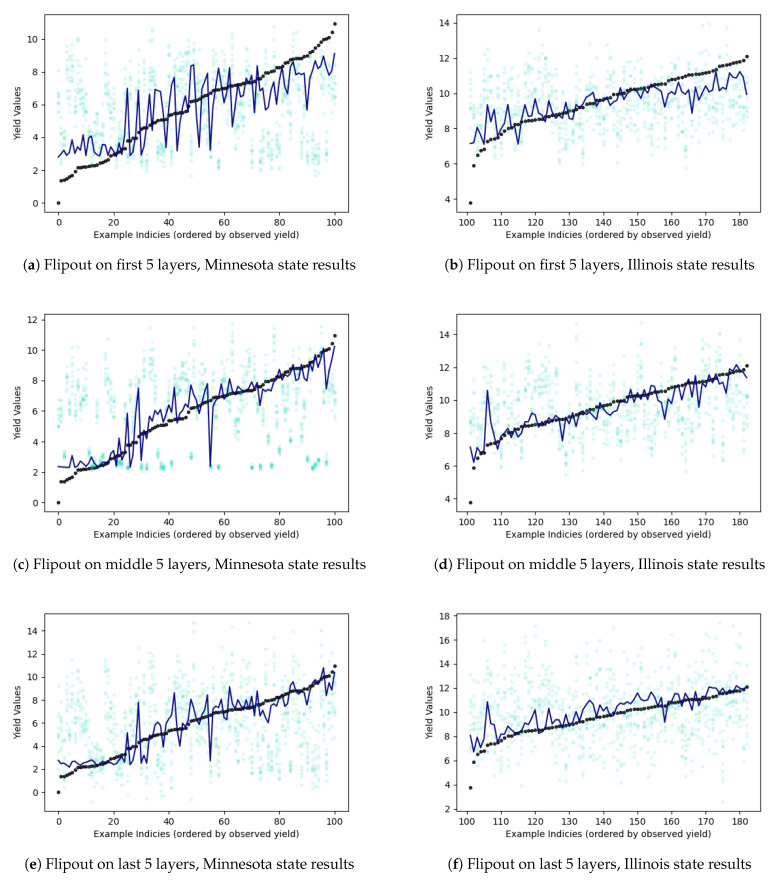
Crop yield predictions. X-axis shows arbitrary county indices which are sorted by the observed yield in ascending order. Y-axis represents the yield value. Black points are the observed yield. Navy blue line is the mean prediction and light blue points are the predictions in several individual runs.

**Figure 2 entropy-23-01546-f002:**
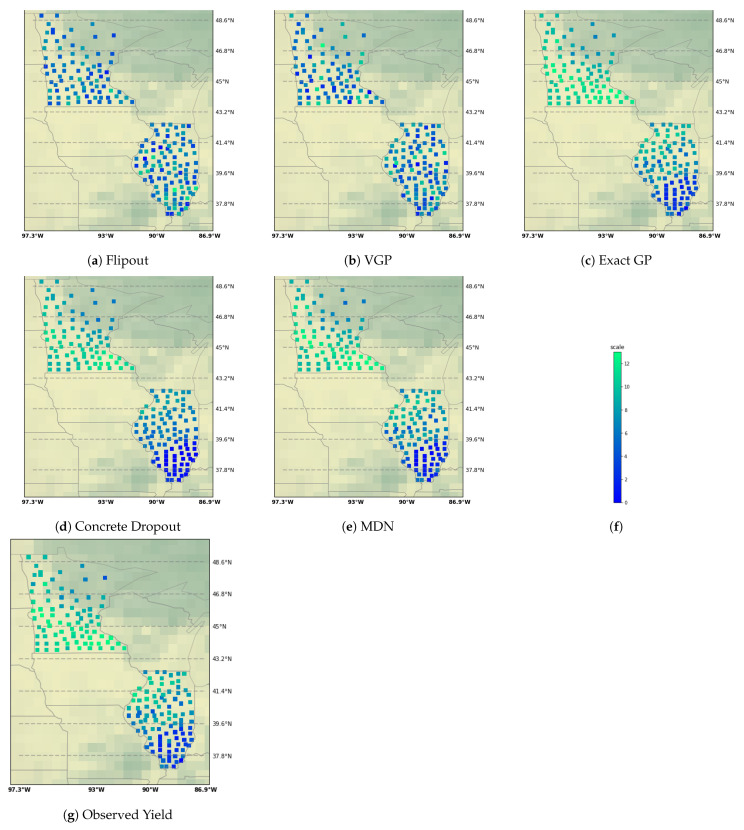
Crop yield predictions for Minnesota and Illinois. Sub-plot (**f**) shows us the legend. Darker blue shade represents lower yield predictions and lighter shade represents higher yield predictions. Methods for better predictive performance (concrete dropout, mixture density network, and exact gp) are able to correctly predict the whole range of observed yield. Flipout and VGP-based Bayesian neural networks are unable to predict well especially in Minnesota counties.

**Figure 3 entropy-23-01546-f003:**
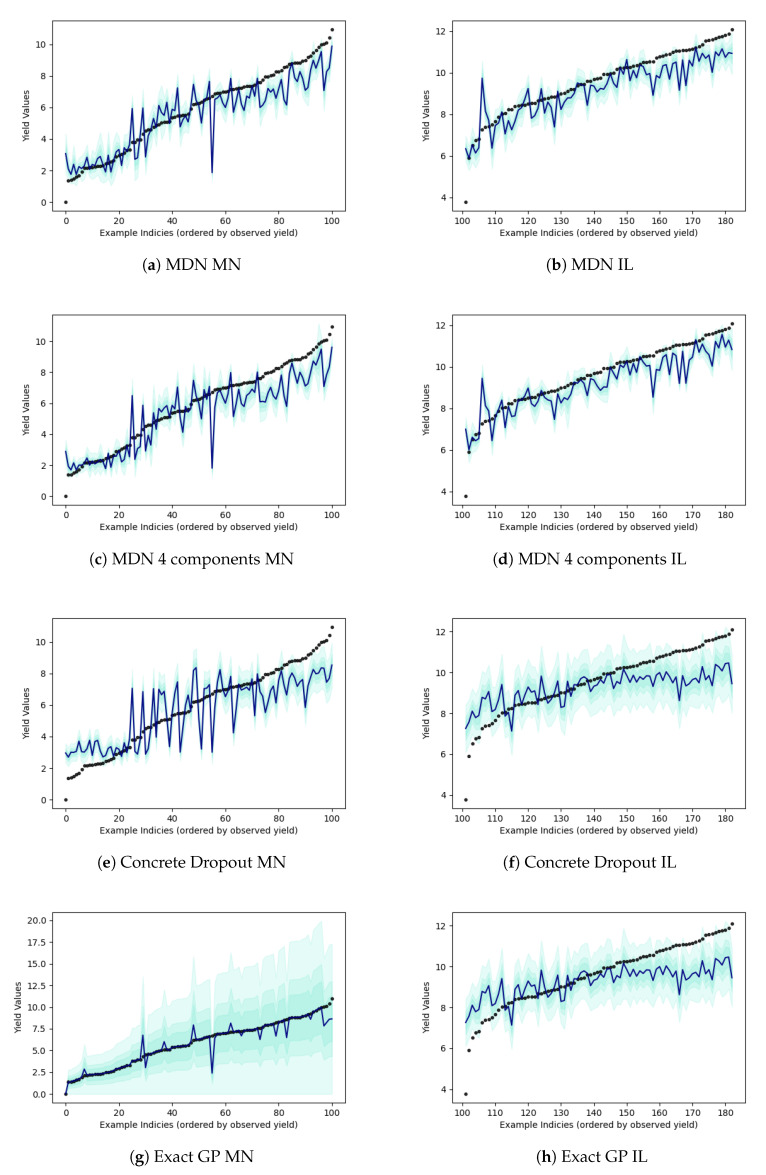
Crop yield predictions. X-axis shows arbitrary county indices which are sorted by the observed yield in ascending order. Y-axis represents the yield value. Black points are the observed yield. Navy blue line is the mean prediction and epistemic uncertainty estimates is shown in turquoise.

**Figure 4 entropy-23-01546-f004:**
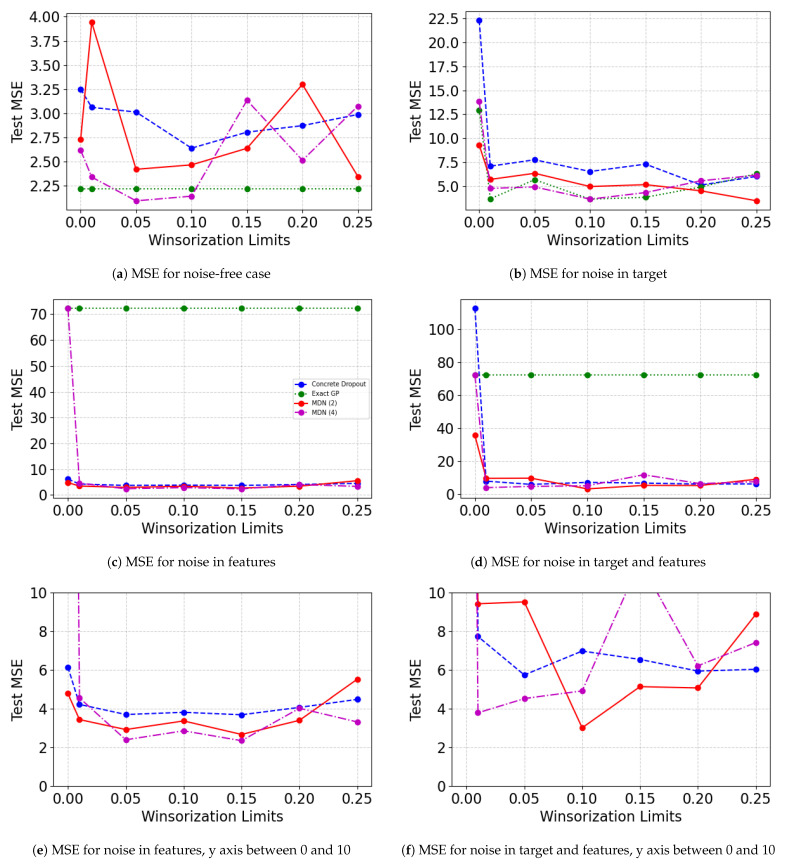
Winsorization results from 0 to 25 percentile limits on crop yield dataset. Mean Squared Error is shown on the y-axis and the Winsorization limits are shown on the x-axis. Different lines represent different methods: Concrete dropout is shown as blue dashed line, exact GP is shown as green dotted line, mixture density network with 2 components is shown in red solid line, and mixture density network with 4 components is shown in magenta dashed-dotted line. As Winsorization limit increases on the training set, the model performance in terms of mean squared error for the untouched test set is shown in the sub-plots.

**Figure 5 entropy-23-01546-f005:**
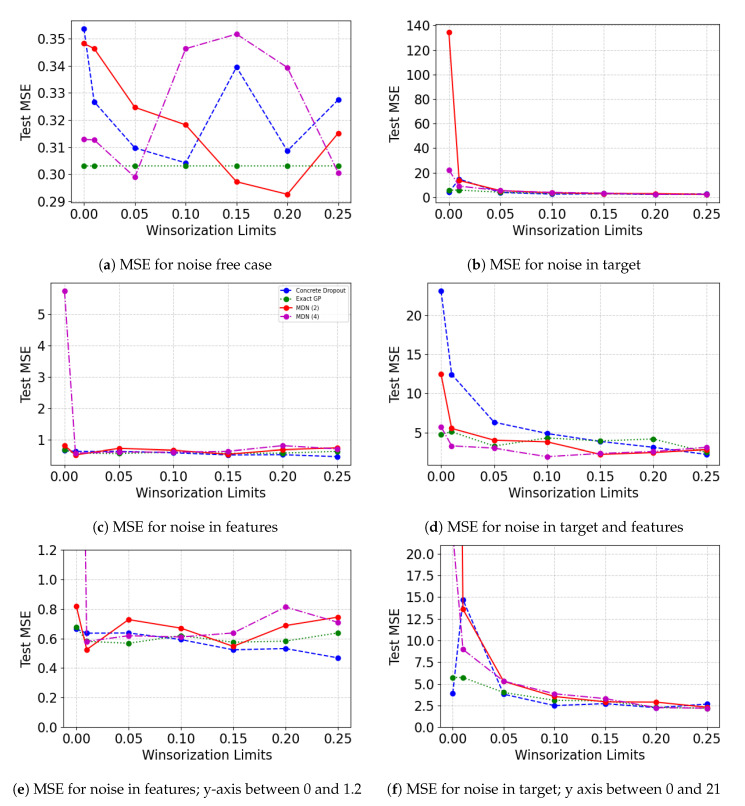
Winsorization results from 0 to 25 percentile limits on California housing dataset. Mean Squared Error is shown on the y-axis and the Winsorization limits are shown on the x-axis. Different lines represent different methods: Concrete dropout is shown as blue dashed line, exact GP is shown as green dotted line, mixture density network with 2 components is shown in red solid line, and mixture density network with 4 components is shown in magenta dashed-dotted line. As Winsorization limit increases in the training dataset, the model performance in terms of mean squared error for the untouched test set is shown in the picture.

**Figure 6 entropy-23-01546-f006:**
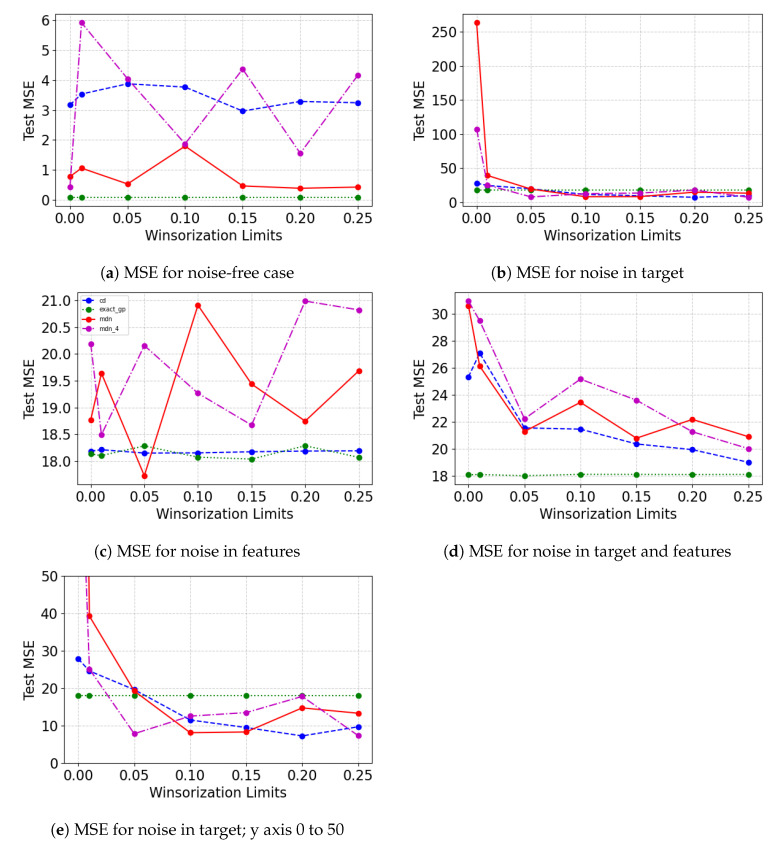
Winsorization results from 0 to 25 percentile limits on Mauna dataset. Mean Squared Error is shown on the y-axis and the Winsorization limits are shown on the x-axis. Different lines represent different methods: Concrete dropout is shown as blue dashed line, exact GP is shown as green dotted line, mixture density network with 2 components is shown in red solid line, and mixture density network with 4 components is shown in magenta dashed-dotted line. As Winsorization limit increases in the training set, the model performance in terms of mean squared error in the untouched test set is shown in the sub-plots.

**Figure 7 entropy-23-01546-f007:**
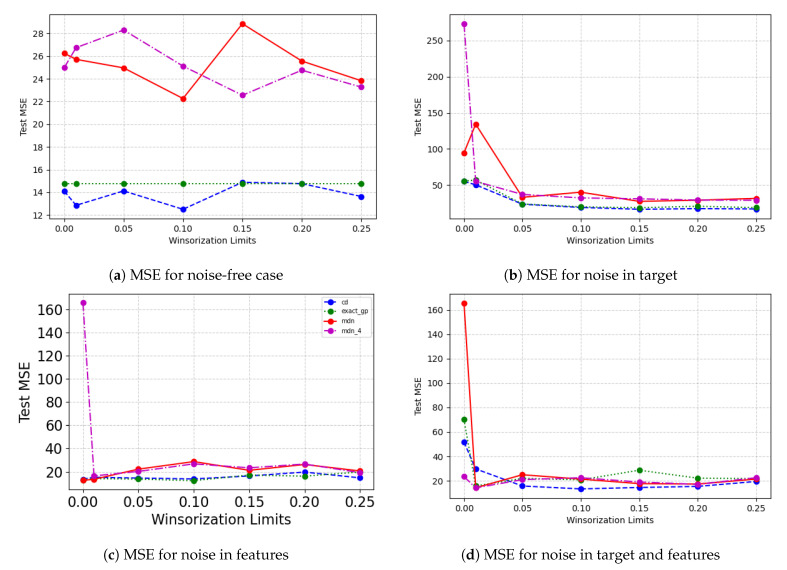
Winsorization results from 0 to 25 percentile limits on forest fires dataset. Mean Squared Error is shown on the y-axis and the Winsorization limits are shown on the x-axis. Different lines represent different methods: Concrete dropout is shown as blue dashed line, exact GP is shown as green dotted line, mixture density network with 2 components is shown in red solid line, and mixture density network with 4 components is shown in magenta dashed-dotted line. As Winsorization limit increases in the training set, the model performance in terms of Mean Squared Error in the untouched test set is show in the sub-plots.

**Figure 8 entropy-23-01546-f008:**
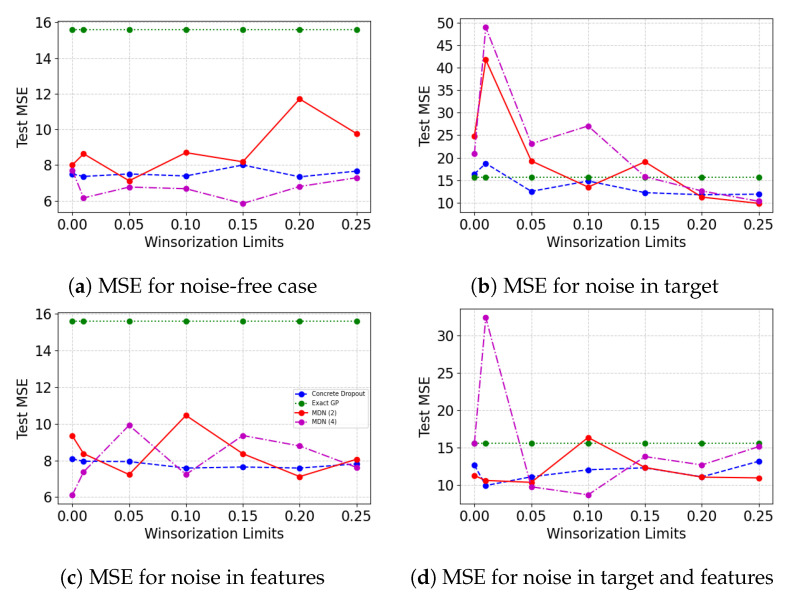
Winsorization results from 0 to 25 percentile limits on GDSC dataset. Mean Squared Error is shown on the y-axis and the Winsorization limits are shown on the x-axis. Different lines represent different methods: Concrete dropout is shown as blue dashed line, exact GP is shown as green dotted line, mixture density network with 2 components is shown in red solid line, and mixture density network with 4 components is shown in magenta dashed-dotted line. As Winsorization limit increases in the training set, the model performance in terms of mean squared error in the untouched test set is shown in the sub-plots.

**Figure 9 entropy-23-01546-f009:**
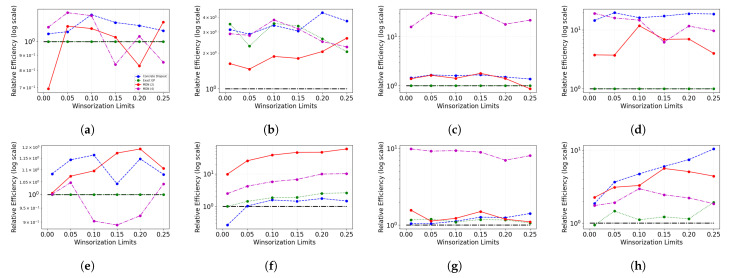
Relative efficiencies (REs) of Winsorized MSE with non-Winsorized MSE for different noise sites. The black dashed line represents an RE of one. RE values greater than one represent improvement in performance with Winsorized training and validation data and vice versa. (**a**) RE for noise free case in crop yield data. (**b**) RE for noise in target in crop yield data. (**c**) RE for noise in features in crop yield data. (**d**) RE for noise in target and features in crop yield data. (**e**) RE for noise free case in California data. (**f**) RE for noise in target in California data. (**g**) RE for noise in features in California data. (**h**) RE for noise in target and features in California data. (**i**) RE for noise free case in GDSC data. (**j**) RE for noise in target in GDSC data. (**k**) RE for noise in features in GDSC data. (**l**) RE for noise in target and features in GDSC data. (**m**) RE for noise free case in forest fires data. (**n**) RE for noise in target in forest fires data. (**o**) RE for noise in features in forest fires data. (**p**) RE for noise in target and features in forest fires data. (**q**) RE for noise free case in Mauna data. (**r**) RE for noise in target in Mauna data. (**s**) RE for noise in features in Mauna data. (**t**) RE for noise in target and features in Mauna data.

**Figure 10 entropy-23-01546-f010:**
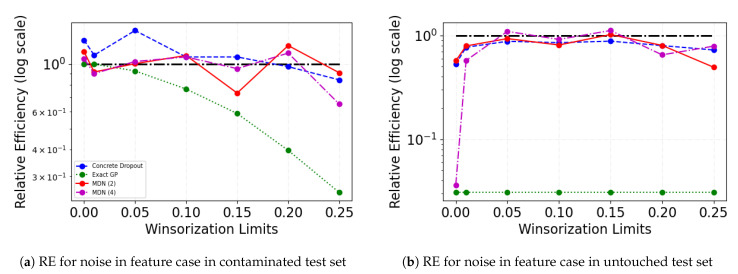
Crop yield dataset result: Relative Efficiencies (RE) comparing performance of Winsorized results with standard Cauchy noise in the features with original performance on noise free data without Winsorization. Black dashed line represents RE of one. REs above one represent improvement in performance due to Winsorization on contaminated data.

**Figure 11 entropy-23-01546-f011:**
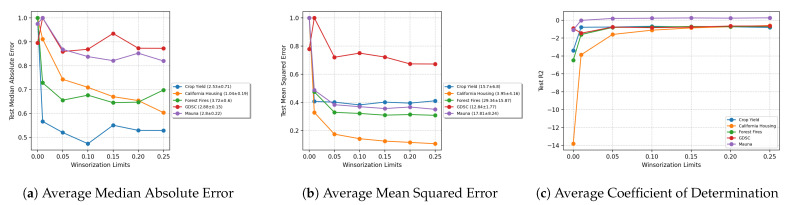
Summarizing Winsorization results: The subplots show average of evaluation metrics over all methodologies used for cases when artificial perturbation is introduced in the datasets. The MSE and Median AE plot legends also convey the mean and standard error of the evaluation metric in the respective sub-plots for each dataset.

**Figure 12 entropy-23-01546-f012:**
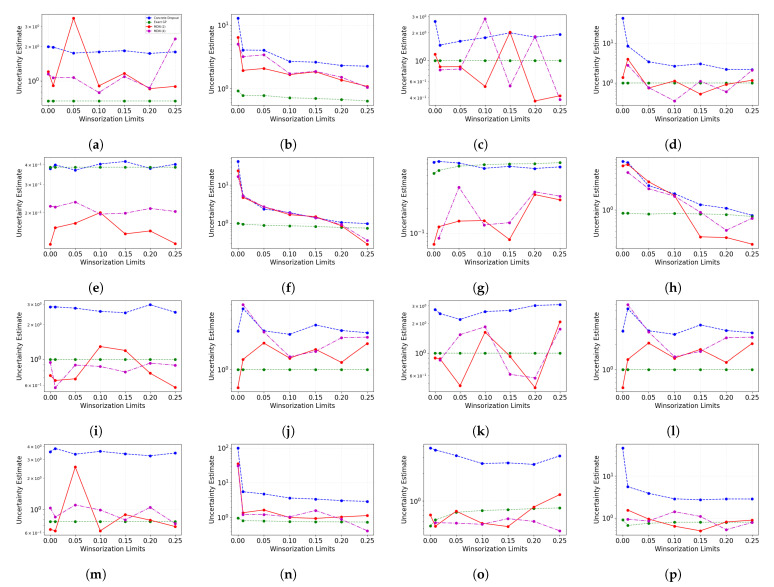
Apart from predictive performance in terms of accurate prediction, the precision can also be compared in terms of uncertainty estimates. On the y-axis, we measure the average standard error in prediction. (**a**) Uncertainty estimate for noise free case in crop yield data. (**b**) Uncertainty estimate for noise in target in crop yield data. (**c**) Uncertainty estimate for noise in features in crop yield data. (**d**) Uncertainty estimate for noise in target and features in crop yield data. (**e**) Uncertainty estimate for noise free case in California data. (**f**) Uncertainty estimate for noise in target in California data. (**g**) Uncertainty estimate for noise in features in California data. (**h**) Uncertainty estimate for noise in target and features in California data. (**i**) Uncertainty estimate for noise free case in GDSC data. (**j**) RE for noise in target in GDSC data. (**k**) Uncertainty estimate for noise in features in GDSC data. (**l**) Uncertainty estimate for noise in target and features in GDSC data. (**m**) Uncertainty estimate for noise free case in forest fires data. (**n**) Uncertainty estimate for noise in target in forest fires data. (**o**) Uncertainty estimate for noise in features in forest fires data. (**p**) Uncertainty estimate for noise in target and features in forest fires data. (**q**) Uncertainty estimate for noise free case in Mauna data. (**r**) Uncertainty estimate for noise in target in Mauna data. (**s**) Uncertainty estimate for noise in features in Mauna data. (**t**) Uncertainty estimate for noise in target and features in Mauna data.

**Table 1 entropy-23-01546-t001:** Comparison of different machine learning models on the test data.

Model	Test MSE	R${}^{2}$
Linear Regression ( Lasso)	2.3432	0.7205
Random Forest	2.1113	0.7481
Support Vector Regression (rbf kernel)	2.3	0.7246
Support Vector Regression (polynomial kernel, degree: 8)	4.2943	0.4878
Concrete Dropout, 3-layer ANN	3.0001	0.6379
Neural Network	1.9224	0.7684

**Table 2 entropy-23-01546-t002:** Probabilistic methods: Crop yield estimation.

Model	Test MSE	R${}^{2}$	Run Time	Test Median Absolute Error	Number of Parameters
Concrete Dropout	2.33	0.62	19 s	1.09	1,106,952
Variation GP	51.09	−7.38	75 s	6.50	1,108,167
Flipout	10,349	−15,544	464 s	73.63	2,213,872
Flipout (early 5 layers)	2.72	0.47	215 s	1.30	1,481,959
Flipout (mid 5 layers)	2.53	0.60	166 s	0.75	1,309,563
Flipout (final 5 layers)	2.70	0.31	232 s	0.85	1,635,316
MDN (2 components)	2.95	0.66	56 s	1.01	1,108,748
MDN (3 components)	2.24	0.69	52 s	0.73	1,110,107
MDN (4 components)	2.15	0.71	54 s	0.71	1,111,466
Exact GP	2.22	0.69	3.6 s	0.69	2

**Table 3 entropy-23-01546-t003:** Winsorization results on test set for crop yield dataset.

Noise Site	Optimal Limit	Model	MSE	MSE${}_{W}$	R${}^{2}$	R${}_{W}^{2}$	Median AE	Median AE${}_{W}$	MAE	MAE${}_{W}$
None	0.1	Concrete Dropout	3.25	2.63	0.55	0.64	1.55	1.16	1.51	1.33
None	0.1	Exact GP	2.21	2.21	0.69	0.69	0.69	0.69	1.13	1.13
None	0.1	MDN	2.72	2.46	0.63	0.66	1.05	0.72	1.28	1.15
None	0.1	MDN (4)	2.61	2.14	0.64	0.70	0.91	0.50	1.19	1.01
*Untouched Test Set*										
Target	0.25	Concrete Dropout	22.27	6.02	−2.02	0.18	4.37	1.55	4.16	1.97
Target	0.25	Exact GP	12.94	6.31	−0.75	0.14	1.74	1.26	2.45	1.89
Target	0.25	MDN	9.29	3.49	−0.26	0.52	1.49	1.40	2.25	1.56
Target	0.25	MDN (4)	13.84	6.15	−0.87	0.17	2.44	1.12	2.91	1.79
Features	0.15	Concrete Dropout	6.15	3.69	0.16	0.50	1.98	1.26	2.07	1.52
Features	0.15	Exact GP	72.27	72.27	−8.79	−8.79	8.6	8.6	8.05	8.05
Features	0.15	MDN	4.78	2.67	0.35	0.63	1.87	0.98	1.86	1.32
Features	0.15	MDN (4)	72.27	2.34	−8.79	0.68	8.60	0.96	8.05	1.21
Target and Features	0.25	Concrete Dropout	112.62	6.03	−14.26	0.18	9.93	1.17	10.33	1.82
Target and Features	0.25	Exact GP	72.27	72.27	−8.79	−8.79	8.60	8.60	8.05	8.05
Target and Features	0.25	MDN	35.55	8.88	−3.81	−0.20	5.44	2.48	5.52	2.46
Target and Features	0.25	MDN (4)	72.27	7.42	−8.79	−0.01	8.60	1.31	8.05	1.98
*Contaminated Test Set*										
Target	0.25	Concrete Dropout	2.85	3.04	0.61	0.58	1.37	1.47	1.40	1.49
Target	0.25	Exact GP	2.21	2.91	0.69	0.60	0.69	1.49	1.13	1.46
Target	0.25	MDN	3.05	2.73	0.58	0.62	1.49	1.69	1.44	1.44
Target	0.25	MDN (4)	2.37	2.93	0.67	0.60	0.81	0.95	1.14	1.32
Features	0.05	Concrete Dropout	2.51	2.27	0.65	0.69	1.19	1.08	1.33	1.17
Features	0.05	Exact GP	2.21	2.39	0.69	0.68	1.13	1.23	0.62	0.61
Features	0.05	MDN	2.39	2.71	0.67	0.63	0.70	1.02	1.09	1.28
Features	0.05	MDN (4)	2.47	2.55	0.66	0.65	1.02	0.84	1.18	1.26
Target and Features	0.25	Concrete Dropout	3.08	3.38	0.58	0.54	1.35	1.64	1.52	1.58
Target and Features	0.25	Exact GP	2.21	5.89	0.69	0.20	0.69	1.98	1.13	2.12
Target and Features	0.25	MDN	2.60	2.09	0.64	0.71	0.87	1.23	1.20	1.20
Target and Features	0.25	MDN (4)	4.12	2.69	0.44	0.63	1.44	1.07	1.63	1.36

**Table 4 entropy-23-01546-t004:** Probabilistic methods: California house prices.

Model	Test MSE	R${}^{2}$	Run Time	Test Median Absolute Error	Number of Parameters
Concrete Dropout	0.44	0.63	28 s	0.26	1,050,168
Variational GP	1.74	−2.39	36 s	0.71	1,050,143
Exact GP	0.28	0.69	73.34 s	0.21	2
Flipout	0.64	−0.54	392 s	0.45	2,100,304
MDN (2 components)	0.31	0.66	52 s	0.23	1,051,964

**Table 5 entropy-23-01546-t005:** Winsorization results on test set for the California Housing dataset.

Noise Site	Optimal Limit	Model	MSE	MSE${}_{W}$	R${}^{2}$	R${}_{W}^{2}$	Median AE	Median AE${}_{W}$	MAE	MAE${}_{W}$
None	0.05	Concrete Dropout	0.35	0.30	0.62	0.67	0.25	0.24	0.38	0.36
None	0.05	Exact GP	0.30	0.30	0.67	0.67	0.20	0.20	0.34	0.34
None	0.05	MDN	0.35	0.32	0.62	0.65	0.23	0.22	0.36	0.35
None	0.05	MDN (4)	0.31	0.29	0.66	0.68	0.18	0.20	0.33	0.34
*Untouched Test Set*										
Target	0.25	Concrete Dropout	3.91	2.65	−3.18	−1.84	0.80	1.58	1.26	1.51
Target	0.25	Exact GP	5.72	2.18	−5.12	−1.33	1.97	1.32	2.18	1.34
Target	0.25	MDN	134.64	2.29	−142.85	−1.44	2.45	1.15	6.89	1.27
Target	0.25	MDN (4)	22.29	2.18	−22.82	−1.33	1.45	1.33	2.56	1.32
Features	0.1	Concrete Dropout	0.66	0.59	0.28	0.36	0.56	0.40	0.64	0.57
Features	0.1	Exact GP	0.67	0.62	0.27	0.33	0.56	0.42	0.62	0.55
Features	0.1	MDN	0.82	0.67	0.12	0.28	0.51	0.45	0.67	0.58
Features	0.1	MDN (4)	5.72	0.61	−5.12	0.34	1.97	0.36	2.18	0.54
Target and Features	0.25	Concrete Dropout	23.10	2.21	−23.68	−1.37	4.55	1.43	4.48	1.33
Target and Features	0.25	Exact GP	4.77	2.47	−4.10	−1.64	2.07	1.53	2.01	1.42
Target and Features	0.25	MDN	12.47	2.82	−12.32	−2.01	2.32	1.31	2.84	1.37
Target and Features	0.25	MDN (4)	5.72	3.09	−5.12	−2.31	1.97	1.18	2.18	1.36
*Contaminated Test Set*										
Target	0.25	Concrete Dropout	0.31	0.60	0.65	0.35	0.23	0.40	0.36	0.55
Target	0.25	Exact GP	0.30	0.57	0.67	0.38	0.20	0.37	0.34	0.51
Target	0.25	MDN	0.32	0.58	0.65	0.37	0.19	0.36	0.33	0.53
Target	0.25	MDN (4)	0.32	0.56	0.65	0.39	0.17	0.35	0.32	0.52
Features	0.01	Concrete Dropout	0.28	0.33	0.69	0.64	0.23	0.27	0.34	0.37
Features	0.01	Exact GP	0.30	0.30	0.67	0.67	0.20	0.20	0.34	0.34
Features	0.01	MDN	0.32	0.27	0.65	0.71	0.20	0.21	0.35	0.33
Features	0.01	MDN (4)	0.33	0.30	0.64	0.67	0.21	0.22	0.35	0.34
Target and Features	0.25	Concrete Dropout	0.30	0.38	0.67	0.59	0.22	0.33	0.34	0.44
Target and Features	0.25	Exact GP	0.30	0.60	0.67	0.35	0.20	0.39	0.34	0.54
Target and Features	0.25	MDN	0.31	1.41	0.66	−0.51	0.21	0.60	0.34	0.80
Target and Features	0.25	MDN (4)	0.35	1.37	0.62	−0.41	0.19	0.63	0.34	0.81

**Table 6 entropy-23-01546-t006:** Probabilistic methods: Mauna CO${}_{2}$ concentration.

Model	Test MSE	R${}^{2}$	Run Time	Test Median Absolute Error
Concrete Dropout	3.24	0.82	25 s	0.99
VGP	18.44	−0.04	3 s	31.45
Exact GP	0.08	0.99	34 s	0.17
MDN	0.42	0.97	23 s	0.38
Flipout	19.68	−0.85	4 s	211.83

**Table 7 entropy-23-01546-t007:** Winsorization results on test set for the Mauna CO${}_{2}$ dataset.

Noise Site	Optimal Limit	Model	MSE	MSE${}_{W}$	R${}^{2}$	R${}_{W}^{2}$	Median AE	Median AE${}_{W}$	MAE	MAE${}_{W}$
None	0.2	Concrete Dropout	3.16	3.28	0.82	0.81	1.03	1.46	1.35	1.52
None	0.2	Exact GP	0.08	0.08	0.99	0.99	0.17	0.17	0.22	0.22
None	0.2	MDN	0.77	0.38	0.95	0.98	0.50	0.31	0.67	0.46
None	0.2	MDN (4)	0.42	1.55	0.97	0.91	0.33	0.57	0.47	0.88
*Untouched Test Set*										
Target	0.25	Concrete Dropout	27.89	9.71	−0.54	0.46	5.24	2.84	4.84	2.80
Target	0.25	Exact GP	18.08	18.08	−0.001	−0.001	3.15	3.15	3.58	3.58
Target	0.25	MDN	263.51	13.34	−13.58	0.26	3.37	3.12	8.24	3.14
Target	0.25	MDN (4)	107.36	7.35	−4.94	0.59	5.59	2.49	6.92	2.43
Features	0.25	Concrete Dropout	18.18	18.19	−0.006	−0.007	2.97	2.95	3.53	3.53
Features	0.25	Exact GP	18.06	18.06	−0.003	−0.0003	3.15	3.15	3.57	3.57
Features	0.25	MDN	18.76	19.68	−0.03	−0.08	3.20	2.98	3.60	3.65
Features	0.25	MDN (4)	20.19	20.82	−0.11	−0.15	2.88	3.09	3.64	3.75
Target & Features	0.25	Concrete Dropout	25.33	18.99	−0.40	−0.05	5.33	4.17	4.61	3.92
Target & Features	0.25	Exact GP	18.06	20.89	−3e−3	−0.002	3.15	3.15	3.57	3.58
Target & Features	0.25	MDN	30.60	20.89	−0.69	−0.15	4.66	4.13	4.72	4.12
Target & Features	0.25	MDN (4)	30.94	20.01	−0.71	−0.11	4.72	3.98	4.75	3.93
*Contaminated Test Set*										
Target	0.05	Concrete Dropout	3.13	2.95	0.82	0.83	0.87	0.98	1.27	1.27
Target	0.05	Exact GP	0.08	0.15	0.99	0.99	0.17	0.17	0.22	0.26
Target	0.05	MDN	0.70	0.82	0.96	0.95	0.48	0.49	0.64	0.68
Target	0.05	MDN (4)	1.77	0.49	0.90	0.97	0.42	0.35	0.84	0.51
Features	0.05	Concrete Dropout	3.46	3.21	0.80	0.82	0.89	1.11	1.34	1.37
Features	0.05	Exact GP	0.08	1.65	0.99	0.90	0.17	0.19	0.22	0.59
Features	0.05	MDN	1.03	1.14	0.94	0.93	0.45	0.49	0.68	0.74
Features	0.05	MDN (4)	0.80	4.21	0.95	0.76	0.47	0.78	0.63	1.33
Target and Features	0.01	Concrete Dropout	3.48	3.51	0.80	0.80	1.18	0.93	1.42	1.36
Target and Features	0.01	Exact GP	0.08	0.13	0.99	0.99	0.17	0.17	0.22	0.24
Target and Features	0.01	MDN	0.44	0.43	0.97	0.98	0.41	0.34	0.50	0.43
Target and Features	0.01	MDN (4)	3.46	6.33	0.80	0.64	0.95	1.56	1.29	1.87

**Table 8 entropy-23-01546-t008:** Probabilistic methods: Forest fires.

Model	Test MSE	R${}^{2}$	Run Time	Test Median Absolute Error
Concrete Dropout	12.93	−0.11	100 s	3.25
Flipout	2015.67	−192.80	1162 s	3.66
VGP	21.93	−0.73	372 s	3.71
Exact GP	14.66	−0.19	4 s	3.46
MDN	21.44	−0.74	95 s	3.52

**Table 9 entropy-23-01546-t009:** Winsorization results on test set for forest fires dataset.

Noise Site	Optimal Limit	Model	MSE	MSE${}_{W}$	R${}^{2}$	R${}_{W}^{2}$	Median AE	Median AE${}_{W}$	MAE	MAE${}_{W}$
None	0.01	Concrete Dropout	14.10	12.85	−0.15	−0.04	3.39	3.25	3.40	3.22
None	0.01	Exact GP	14.75	14.75	−0.2	−0.20	3.51	3.51	3.39	3.39
None	0.01	MDN	26.24	25.71	−1.14	−1.09	3.76	3.57	4.28	4.03
None	0.01	MDN (4)	24.97	26.74	−1.03	−1.18	3.91	3.83	4.05	4.17
*Untouched Test Set*										
Target	0.25	Concrete Dropout	55.19	16.95	−3.51	−0.38	4.86	2.96	5.98	3.51
Target	0.25	Exact GP	55.73	18.78	−3.55	−0.53	3.84	3.04	5.34	3.66
Target	0.25	MDN	94.78	31.60	−6.73	−1.57	5.27	3.99	6.99	4.70
Target	0.25	MDN (4)	273.36	29.05	−21.31	−1.37	5.16	4.21	8.55	4.62
Features	0.15	Concrete Dropout	13.61	16.39	−0.11	−0.33	2.97	2.68	3.41	3.31
Features	0.15	Exact GP	12.56	17.15	−0.02	−0.40	3.16	3.51	3.29	3.62
Features	0.15	MDN	12.59	21.26	−0.02	−0.73	3.23	3.39	3.08	3.75
Features	0.15	MDN (4)	165.53	23.41	−12.51	−0.91	11.18	3.56	12.38	3.91
Target and Features	0.25	Concrete Dropout	51.78	19.42	−3.22	−0.58	6.57	3.40	6.04	3.77
Target and Features	0.25	Exact GP	70.21	21.73	−4.73	−0.77	6.55	3.95	7.06	8.06
Target and Features	0.25	MDN	165.53	21.39	−12.51	−0.74	11.18	3.42	12.38	3.71
Target and Features	0.25	MDN (4)	23.44	22.49	−0.91	−0.65	3.89	3.55	4.01	3.81
*Contaminated Test Set*										
Target	0.25	Concrete Dropout	13.68	12.95	−0.11	−0.05	3.33	3.04	3.36	3.23
Target	0.25	Exact GP	14.75	14.91	−0.20	−0.21	3.50	3.41	3.39	3.45
Target	0.25	MDN	23.89	14.91	−0.95	−0.84	3.72	3.41	4.06	3.45
Target	0.25	MDN (4)	22.49	20.38	−0.83	−0.66	3.74	3.11	3.93	3.68
Features	0.05	Concrete Dropout	14.23	14.71	−0.16	−0.20	3.10	3.16	3.40	3.37
Features	0.05	Exact GP	14.75	15.80	−0.65	−0.29	3.50	3.43	3.39	3.45
Features	0.05	MDN	20.30	23.54	−0.65	−0.92	3.05	3.72	3.64	4.02
Features	0.05	MDN (4)	27.00	29.38	−1.20	−1.39	4.01	3.85	4.25	4.40
Target and Features	0.25	Concrete Dropout	13.78	46.52	−0.12	−2.79	3.29	5.17	3.37	5.73
Target and Features	0.25	Exact GP	14.75	39.99	−0.20	−2.26	3.50	4.08	3.39	4.08
Target and Features	0.25	MDN	25.54	202.67	−1.08	−15.54	3.85	7.58	4.21	10.52
Target and Features	0.25	MDN (4)	21.82	221.33	−0.78	−17.06	3.91	8.15	3.94	11.31

**Table 10 entropy-23-01546-t010:** Probabilistic methods: GDSC.

Model	Test MSE	R${}^{2}$	Run Time	Test Median Absolute Error
Concrete Dropout	7.77	−0.10	32 s	1.46
Exact GP	15.59	−1.25	3.8 s	3.75
MDN	7.78	−0.13	119 s	1.26
Flipout	80.67	−26.07	970 s	5.95
VGP	6.15	0.08	295 s	1.46

**Table 11 entropy-23-01546-t011:** Winsorization results on test set for the GDSC dataset.

Noise Site	Optimal Limit	Model	MSE	MSE${}_{W}$	R${}^{2}$	R${}_{W}^{2}$	Median AE	Median AE${}_{W}$	MAE	MAE${}_{W}$
None	0.05	Concrete Dropout	7.49	7.49	−0.08	−0.08	2.57	2.35	2.36	2.40
None	0.05	Exact GP	15.59	15.59	−1.25	−1.25	3.75	3.75	3.40	3.40
None	0.05	MDN	7.99	7.11	−0.15	−0.02	1.64	1.60	2.12	2.11
None	0.05	MDN (4)	7.71	6.76	−0.11	0.02	1.73	1.62	2.06	1.98
*Untouched Test Set*										
Target	0.25	Concrete Dropout	16.38	11.89	−1.37	−0.72	3.88	3.51	3.48	2.97
Target	0.25	Exact GP	15.59	15.59	−1.25	−1.25	3.75	3.75	3.40	3.40
Target	0.25	MDN	24.82	9.85	−2.59	−0.42	2.08	2.03	3.36	2.43
Target	0.25	MDN (4)	20.86	10.32	−2.01	−0.49	2.91	2.80	3.58	2.79
Features	0.2	Concrete Dropout	8.08	7.58	−0.16	−0.09	2.46	2.50	2.51	2.45
Features	0.2	Exact GP	15.59	15.59	−1.25	−1.25	3.75	3.75	3.40	3.40
Features	0.2	MDN	9.34	7.11	−0.35	−0.03	1.44	1.95	2.33	2.22
Features	0.2	MDN (4)	6.11	8.78	0.11	−0.27	1.57	2.26	2.00	2.54
Target and Features	0.25	Concrete Dropout	12.65	13.18	−0.83	−0.90	3.54	3.69	3.08	3.15
Target and Features	0.25	Exact GP	15.59	15.59	−1.25	−1.25	3.75	3.75	3.40	3.40
Target and Features	0.25	MDN	11.26	10.95	−0.63	−0.58	3.25	3.28	2.96	2.69
Target and Features	0.25	MDN (4)	15.59	15.13	−1.25	−1.19	3.75	3.51	3.40	3.37
*Contaminated Test Set*										
Target	0.25	Concrete Dropout	7.55	5.99	−0.09	0.13	2.31	2.19	2.32	2.12
Target	0.25	Exact GP	15.59	15.59	−1.25	−1.25	3.75	3.75	3.40	3.40
Target	0.25	MDN	12.23	6.67	−0.77	0.03	2.14	1.68	2.78	2.08
Target	0.25	MDN (4)	5.96	6.36	0.13	0.07	1.49	1.93	1.84	2.16
Features	0.25	Concrete Dropout	7.79	7.61	−0.12	−0.10	2.52	2.43	2.40	2.41
Features	0.25	Exact GP	15.59	8.70	−1.25	−0.26	3.75	2.55	3.40	2.54
Features	0.25	MDN	7.99	8.14	−0.15	−0.17	2.08	1.99	2.37	2.34
Features	0.25	MDN (4)	10.77	20.3	−0.55	−1.94	2.09	2.42	2.77	3.52
Target and Features	0.25	Concrete Dropout	7.22	6.71	−0.04	0.02	2.41	1.95	2.33	2.22
Target and Features	0.25	Exact GP	15.59	7.18	−1.25	−0.03	3.75	2.29	3.4	2.38
Target and Features	0.25	MDN	5.24	7.45	0.24	−0.07	1.76	1.62	1.95	2.11
Target and Features	0.25	MDN (4)	9.03	7.32	−0.30	−0.05	1.95	2.53	2.44	2.40

## Data Availability

We have used publicly available datasets in this paper.

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
