# Peer review of "Winsorization for Robust Bayesian Neural Networks"

_entropy, 2021, doi:10.3390/e23111546_

Round 1

Reviewer 1 Report

The authors counducted a very thorough study of robustness of various machine learning models, both deterministic and probabilistic, when the outliers in training data were trimmed based on the Winsorization method. From the extensive results, the mixture density network perform well with or without trimming the outliers. GP, which is single Gaussian, is reasonably sensitive to the presence or absence of outliers. Although the paper does not propose new models, the thorough experiments on the real data using various models is a plus. I recommend publication of the paper.

Two minor points that the authors may consider before the final version.

    1. In eq.12, the log term in the reparameterization should be square root of log so one can reproduce the covariance of log(1+Sigma). The authors should double check it.
    2. In a talk by Neil Lawrence, http://inverseprobability.com/talks/notes/deep-gaussian-processes.html, he mentioned he used deep GP model to fit a marathon data, finding that the DGP model is not susceptible to the presence of a single outlier. Many recent papers on DGP mentioned the heavy-tailed aspect of deep GP, e.g. Duvenaud, et. al. "Avoiding pathology in very deep neural network" AISTATS 2014; Lu, et. al. "Interpretable deep Gaussian Process with moments" AISTATS 2020; Pleiss and Cunningham arXiv:2106.06529.

Author Response

We are grateful to the reviewers for taking time to provide valuable feedback on our work. The individual comments are displayed in italics and the response is displayed in regular font. Changes to the manuscript have been noted in these responses.

  • Comment 1: In eq.12, the log term in the reparameterization should be square root of log so one can reproduce the covariance of log(1+Sigma). The authors should double check it.

We have followed the framework of Blundell et al (2015), where $log(1+ exp(\Sigma)$ is the standard deviation (and not the variance). Consequently, we have made the appropriate change at line 218 before equation 12, and this now reads ``...$\mathbf{W}_{ij}$ is drawn from $\mathcal{N}(\mu_{ij}, log(1+ exp(\Sigma_{ij}))^2)$...''.  We thank the reviewer for pointing out this issue. 

  • Comment 2: In a talk by Neil Lawrence, http://inverseprobability.com/talks/notes/deep-gaussian-processes.html, he mentioned he used deep GP model to fit a marathon data, finding that the DGP model is not susceptible to the presence of a single outlier. Many recent papers on DGP mentioned the heavy-tailed aspect of deep GP, e.g. Duvenaud, et. al. "Avoiding pathology in very deep neural network" AISTATS 2014; Lu, et. al. "Interpretable deep Gaussian Process with moments" AISTATS 2020; Pleiss and Cunningham arXiv:2106.06529.

We thank the referee for pointing out this interesting connection, this clearly deserves further study. While we have not explicitly used deep Gaussian Process here, methods like concrete dropout try to approximate Deep Gaussian Processes. It would be interesting to study if the heavy-tailed behavior is emulated by concrete dropout and similar approximation methods. Text has been added at line 678 in the Conclusion section to reflect our intent to explore this in future work.

Reviewer 2 Report

The authors presented a very large and voluminous work on a relevant topic.
The article makes a good impression.
 However, there are a number of comments.
The article uses abbreviations. All of them must be deciphered. For example, what is NLP on line 88? We may know this, but it would be nice to indicate the decryption.
The same remark applies to formulas. A description of the symbols used should be added to all formulas. For example, in formula (1), the authors use E., it is necessary to explain this meaning in the text (although most interested readers will understand the meaning of this designation).
Inserting explanations for formulas will improve the overall impression of the work.

There are typos in the article. What does "whoise" mean on the last line on page 4?
Should the word "Foctorization" be corrected on line 197?

What do authors think about the materials of this article: "Tarasov, IE A Mathematical Method for Determining the Parameters of Functional Dependencies Using Multiscale Probability Distribution Functions. Mathematics 2021, 9, 1085. https://doi.org/10.3390/math9101085", which discusses the application of the method of approximation of experimental data by functional dependencies, which uses a probabilistic assessment of the deviation of the assumed dependence from experimental data?

In general, the article can be published after the edits made by the authors.

Author Response

We are grateful to the reviewers for taking time to provide valuable feedback on our work. The individual comments are displayed in italics and the response is displayed in regular font. Changes to the manuscript have been noted in these responses.

  • Comment 1: The article uses abbreviations. All of them must be deciphered. For example, what is NLP on line 88? We may know this, but it would be nice to indicate the decryption.

The same remark applies to formulas. A description of the symbols used should be added to all formulas. For example, in formula (1), the authors use E., it is necessary to explain this meaning in the text (although most interested readers will understand the meaning of this designation).

Inserting explanations for formulas will improve the overall impression of the work.

There are typos in the article. What does "whoise" mean on the last line on page 4?

Should the word "Foctorization" be corrected on line 197?

We appreciate the reviewer's comments regarding typos, abbreviations, and formulas. We have made the necessary changes to include full form of the words before using the abbreviations, to explain terms in formulas when they first occur and we have remove all the typos that we found. These changes are reflected on line 88, 89, 116, 169, 185, 198 and 252.

  • Comment 2: What do authors think about the materials of this article: "Tarasov, IE A Mathematical Method for Determining the Parameters of Functional Dependencies Using Multiscale Probability Distribution Functions. Mathematics 2021, 9, 1085. https://doi.org/10.3390/math9101085", which discusses the application of the method of approximation of experimental data by functional dependencies, which uses a probabilistic assessment of the deviation of the assumed dependence from experimental data?

This is an interesting article, we thank the referee for bringing this to our attention. We have added some text in line 739 to reflect the possibility of future developments using Winsorization and other methods developed in this paper in the light of this article.